**Strengthening the biogeosciences in environmental research networks**

Daniel D. Richter[1], Sharon A. Billings[2], Peter M. Groffman[3], Eugene F. Kelly[4], Kathleen A. Lohse[5], William H. McDowell[6], Timothy S. White[7], Suzanne Anderson[8], Dennis D. Baldocchi[9], Steve Banwart[10], Susan Brantley[11], Jean J. Braun[12], Zachary S. Brecheisen[1], Charles W. Cook[1], Hilairy E. Hartnett[13], Sarah E. Hobbie[14], Jerome Gaillardet[15], Esteban Jobbagy[16], Hermann F. Jungkunst[17], Clare E. Kazanski[18], Jagdish Krishnaswamy[19], Daniel Markewitz[20], Katherine O'Neill[21], Clifford S. Riebe[22], Paul Schroeder[23], Christina Siebe[24], Whendee L. Silver[25], Aaron Thompson[26], Anne Verhoef[27], Ganlin Zhang[28]

[1] Nicholas School of the Environment, Duke University, Durham, NC 27708 USA
[2] Department of Ecology and Evolutionary Biology, Kansas Biological Survey, University of Kansas, Lawrence, KS USA
[3] City University of New York, Advanced Science Research Center and Brooklyn College, Department of Earth & Environmental Sciences, New York, NY USA
[4] Department of Soil and Crop Sciences, Colorado State University, Ft. Collins, CO USA
[5] Department of Biological Sciences, Idaho State University, Pocatello, ID USA
[6] Department of Natural Resources and the Environment, University of New Hampshire, Durham, NH USA
[7] Earth and Environmental Systems Institute, The Pennsylvania State University, University Park, PA USA
[8] Institute of Arctic and Alpine Research and Dept. of Geography, University of Colorado, Boulder, CO USA
[9] Environmental Science, Policy, and Management, University of California at Berkeley, CA USA
[10] School of Earth and Environment, University of Leeds, UK
[11] Earth and Environmental Systems Institute, The Pennsylvania State University, University Park, PA USA
[12] Geosciences Environment Toulouse, Universite de Toulouse, Toulouse, FR and University of Yaounide, LIM DYCOFAC, IRD, Yaounde, Cameroon
[13] School of Earth and Space Exploration and School of Molecular Sciences, Arizona State University, Tempe, AZ USA
[14] Department of Ecology, Evolution, and Behavior, University of Minnesota, St. Paul, MN USA
[15] Institut de Physique du Globe de Paris, Institut Universitaire de France, Paris, France
[16] Grupo de Estudios Ambientales - IMASL, CONICET, and Universidad Nacional de San Luis, Argentina
[17] Institute for Environmental Sciences, University of Koblenz-Landau, Landau, Germany
[18] Department of Ecology, Evolution, and Behavior, University of Minnesota, St. Paul, MN, USA
[19] Ashoka Trust for Research in Ecology and the Environment (ATREE), Bangalore, India
[20] Warnell School of Forestry and Natural Resources, University of Georgia, Athens, GA USA
[21] Environmental Studies, Roanoke College, Salem VA USA
[22] Department of Geology and Geophysics, University of Wyoming, Laramie, WY USA
[23] Department of Geology, University of Georgia, Athens, GA USA
[24] Instituto de Geologia, Universidad Nacional Autonoma de Mexico, Mexico City, MX
[25] Department of Environmental Science, Policy, and Management, University of California, Berkeley, CA USA
[26] Department of Crop and Soil Sciences, University of Georgia, Athens, GA USA
[27] Department of Geography and Environmental Science, The University of Reading, Reading, UK

[28] State Key Laboratory of Soil and Sustainable Agriculture, Institute of Soil Science, Chinese Academy of Sciences, Nanjing 210008, China

*Correspondence to:* Daniel D. Richter (drichter@duke.edu)

**Strengthening the biogeosciences in environmental research networks**


       Many scientific approaches are improving our understanding and management of the rapidly changing environment. Long-term environmental research networks are one approach to advancing local, regional, and global environmental science and education. A remarkable number and wide variety of environmental research networks operate around the world today.

These are diverse in funding, infrastructure, motivating questions, scientific strengths, and the sciences that birthed and maintained the networks. Some networks have individual sites that were selected because they had produced invaluable long-term data, while other networks have new sites selected to span ecological gradients. However, all long-term environmental networks share two challenges. Networks must keep pace with scientific advances and interact with both

the scientific community and society at large. If networks fall short of successfully addressing these challenges, they risk becoming irrelevant. The objective of this paper is to assert that the biogeosciences offer environmental research networks a number of opportunities to expand scientific impact and public engagement. We explore some of these opportunities with four networks: the International Long Term Ecological Research programs (ILTERs), the Critical

Zone Observatories (CZOs), the Earth and Ecological Observatory networks (EONs), and the FLUXNET program of eddy flux sites. While these networks were founded and grown by interdisciplinary scientists, the preponderance of expertise and funding have gravitated activities of ILTERs and EONs toward ecology and biology, CZOs toward the Earth sciences and geology, and FLUXNET toward ecophysiology and micrometeorology. Our point is not to

homogenize networks, nor to diminish disciplinary science. Rather, we argue that by more fully incorporating the integration of biology and geology in long-term environmental research networks, scientists can better leverage network assets, keep pace with the ever-changing science of the environment, and engage with larger scientific and public audiences.


## 1. Introduction

In this paper, we bring the biogeosciences and environmental research networks together by exploring their origins and by asking a simple question: might on-going environmental research networks benefit from a perspective that more explicitly includes the biogeosciences? The specific objectives of this paper are to consider the historical development of the biogeosciences and of environmental research networks, and to use that history to highlight opportunities for the world's environmental research networks to use the biogeosciences to benefit network science itself and to broaden their impacts on the wider sciences and society.

Growing numbers of biologists and geologists are working together on the biogeoscience of societally important issues (Hedin et al., 2002; Hinckley et al., 2016; Field et al., 2016; O'Neill and Richter, 2016; Wymore et al., 2017; Brantley et al., 2017a). Top-tier, multidisciplinary journals now publish biogeoscience papers, and professional ecological and geological societies have new biogeoscience journals and subdivisions. The highly cited and venerable journal *Biogeochemistry* has been in publication since 1984. New biogeoscience awards and lectureships are funded. Cambridge University Press recently published a major volume entitled, *A Biogeoscience Approach to Ecosystems* (Johnson and Martin, 2016). We write this paper to assert that there is scientific potential to bringing a biogeoscience-explicit perspective to the world's environmental research networks, and that biogeoscience initiatives at individual sites or across networks can increase the value of environmental research networks for science, education, and society at large.

## 2. Biogeoscience past and present

To consider the origins of biogeoscience and thereby develop a perspective for its further application to environmental research networks, we must mention the incomparable biogeoscientist, Alexander von Humboldt, widely recognized as the founder of biogeography. But we begin with some detail with Darwin, whose evolutionary biology is deeply seated in biogeoscience. For this Darwin owes much to Lyell, whose *Principles of Geology* opened for the young Darwin the geologic history of the Earth, as an ancient, life-filled, and highly dynamic planet. Lyell's three-volume *Principles* were among Darwin's most important books in the *Beagle's* 400-book library (Herbert, 2005). After the *Beagle's* five-year voyage around the world, Darwin's *Voyages* and *The Structure and Distribution of Coral Reefs* both vigorously embraced geology *and* biology. Chancellor (2008) described *Coral Reefs* as "not just a book about reefs, it is a book which sweeps across the ecology and geology of the whole world." In 1859, Darwin

was awarded the Wollaston Medal by the Geological Society of London, the geological society's highest award. Darwin spoke spiritedly about the concert of geology and biology; "My books came half out of Lyell's brains," he wrote to a colleague (CD Letter to Leonard Horner, 8-29-

1844). Darwin's genius sprang from his understanding that the overlap and interaction of biology and geology drove biological evolution.

By the early 20[th] century, however, biology and geology had subdivided into distinct disciplines. The growing recognition of biological and geological complexities facilitated the division, but so too did scientific reductionism and the departmentalization of university faculties.

Many academics welcomed the narrowing of scope and the close-knit academic communities brought about by departments (Stichweh, 1992). Despite exceptions, even today the mainstreams of biology and geology remain formally separated by university departments, professional societies, funding streams, and journals. With the exception of paleontology, geomicrobiology, and evolutionary theory, their vast literatures rarely reference the other.

Despite the formal division, the two sciences have been bridged by a number of remarkable biogeoscientists. Darwin champion, Thomas H. Huxley, lectured for many decades not only on the veracity of biological evolution but also on the close relations and interactions of biology and geology. In a public lecture Huxley (1897) called "one of the greatest chapters in the history of the world," he told a story he said was written in a simple piece of chalk. Huxley began

by remarking that if a chalk rock is viewed under a microscope, it is seen to be a collection of the most beautiful tiny shells of a fossil organism named *Globigerina*. "A cubic inch may contain a hundred thousand of their bodies". After noting that *Globigerina* was but one of the fossils, and how ancient these fossils were, about a 100 million years old, he then celebrated the then recent discovery of living *Globigerina*, a discovery made during the laying of the telegraph cable

on the ocean bottom between Ireland and Newfoundland. As the cable rested on the bottom of the Atlantic, the ocean's depth had to be measured and the seafloor sampled over several 1000s of miles. Most of the seafloor, Huxley exclaimed, was discovered to be beds of recently deceased *Globigerina* and similar creatures that had died and accumulated on the bottom of the ocean and making an ideal surface for the cable. Huxley's story is about life in a rock and it

spans the microscopic to the vast ocean, and the present with the many millions of years. He emphatically claimed that such stories are fundamental to the education of well educated scholars and the general public.

Any history of biogeoscience must include the Ukrainian-Russian Vladimir Vernadsky (1998, originally for 1926), a mineralogist by training, whose most important book, *The*

*Biosphere*, introduced the new science of biogeochemistry. Vernadsky saw the Earth as a

dynamic planet driven by tightly linked biogeochemical reactions of photosynthesis, decomposition, chemical cycling, and mineral weathering. Figure 1, a recent satellite image of Earth's photosynthetic activity, presents a vision that is Vernadsky's, of Earth as a metabolic system. Not widely known is that one of Vernadsky's most influential teachers was the

renowned Vasily Dokuchaev, an inspirational teacher widely recognized to be founder of the biogeoscience known as pedology (Jenny, 1961).

Vernadsky's *Biosphere* though quickly translated into French, went untranslated into English for many decades. Vernadsky's ideas however circulated within the English-speaking world in part due to their active promotion by the ecologist G.E Hutchinson. Hutchinson urged

his students and fellow scientists to "confront all of the processes that maintain or change ecological systems, whether these processes were biological, physical or geological" (Slobodkin, 1993). Hutchinson immediately made use of Tansley's (1935) coinage of the ecosystem concept, i.e., the indivisible physical system of biota and environment. He was also instrumental in helping the young Ray Lindeman (1942) with his mathematical models of the

biogeochemical cycles of a lake ecosystem. Hutchinson famously intervened to advocate for the publication of Lindeman's (1942) paper, in which Lindeman wrote that the "constant organic–inorganic cycle of nutritive substance is so completely integrated that to consider … a lake primarily as a biotic community appears to force a "biological" emphasis upon a more basic functional organization."  Hutchinson and his students brought an expansive sense of space and

time to ecosystem science, coring many tens of meters into lake sediments to reconstruct the multi-millennial evolution of lakes and of their surrounding catchments (Hutchinson and Wollack, 1940).

By the time that Hutchinson was writing "The Biosphere" for *Scientific American* (1970), an essay all but formally dedicated to Vernadsky, the International Biological Program (IBP) was

systematically gathering enormous amounts of ecosystem data from tropical forests to the tundra. The IBP represents one of the world's first comprehensive environmental and ecosystem research networks, complete with standardized protocols and data management. Remarkably, IBP research was funded by many nations irrespective of politics, and by the early 1980s, the IBP had assembled a vast collection of new biogeochemistry data from hundreds of

sites (e.g., Reichle, 1981).  The IBP program greatly accelerated our understanding of ecosystems at local to global scale and helped spread the concept of the ecosystem, what some have called the "biogeocenoesis", worldwide.

Two additional historical developments, those of the watershed ecosystem and the critical zone ecosystem, pertain directly to the relations between biogeoscience and environmental research networks.

First, hydrologists have quantified how streamflow responds to precipitation, how land management alters watershed response, and how evapotranspiration varies as a function of water supply and evaporative demand. Watershed experiments have long been conducted internationally in developed and developing nations (e.g., Hursh et al., 1942; Krishnaswamy, 2017), and watershed monitoring and models have grown ever more sophisticated (Figure 2). Bormann (1996) described how these watershed studies led directly to the measurement of chemical element inputs and outputs in precipitation and stream water, respectively, and to the birth of the watershed ecosystem concept (Bormann and Likens, 1967). Watersheds not only control hydrologic responses but also rates of weathering, erosion, and biogeochemical cycling of chemical elements. The science of watershed ecosystems is nothing if not a biogeoscience.

Related to the concept of the watershed ecosystem is the concept of the critical zone ecosystem. In 2001, a group of Earth scientists and ecologists proposed the Earth's critical zone as a concept that integrates the structure and interconnected dynamics of the atmosphere, the vegetation, soils, and underlying regolith down to the deepest groundwater and weathering fronts (National Research Council, 2001). Critical zone science was proposed as a new Earth system science, an explicitly interdisciplinary and integrative science that includes all Earth systems sciences. Given the usefulness of Bormann and Likens' concept of watershed ecosystem, we propose greater use of "critical zone ecosystem" to draw greater attention to all boundaries of Earth's life support system, but specifically to those that are subsurface and generally considered "geological". The critical zone ecosystem operates and evolves across time scales from the instantaneous to the multi-million years, and like Tansley's ecosystem and the watershed ecosystem, the critical zone ecosystem is spatially scalable (Evans, 1956) from vegetation-clad soil and regolith profiles, to small watersheds and large river basins, to the continental and the global terrestrial surface (Richter and Billings, 2015). To date, the critical zone ecosystem has received attention primarily via its components, through researchers focused on specific parts of the larger system.

## 3. Origins of environmental research networks

The origins of environmental research networks can be traced to place-based research studies of the 19[th] century that were motivated by famines and rising concerns that farming might not be able to provide sufficient food for growing human populations (Richter and

Markewitz, 2001). Long-term agricultural experiments were initiated, motivated by the prospects that agricultural science might increase and sustain crop yields (Rossiter, 1975). Examples include the Park Grass experiment in England, begun in 1856 to quantify how hayfields respond

to soil amendments (an experiment that Tilman et al. (1994) called the world's "most long-term ecological study"), and the Lethbridge and Breton Plots in Alberta, Canada (established 1910 and 1930, respectively) to test the conversion of native prairie grasslands to cultivation-based agriculture and rotations (McGill et al., 1986). Long-term agricultural field studies spread to the developing world, for example, to China, India, and Pakistan (Tirol-Padre and Ladha, 2006),

where today many dozens of long-term field experiments are used by scientists to test relationships between soil, management, and yields in intensively managed rice, *Oryza* spp. (Bhandari et al., 2002; Tirol-Padre and Ladha, 2006).

Based on a recent international inventory (http://iscn.fluxdata.org/partner-networks/long-term-soil-experiments/), there are many 100s of long-term agricultural research sites world-wide

that are monitoring the sustainability of agricultural production over decadal time scales (Richter and Yaalon, 2012). These experiments study effects of tillage practices, rotations, and long-term amendments of fertilizers and organic materials such as manures and sludges on soils, microbial communities, biochemical and physical fluxes (such as those affecting soil water- and heat regimes), and crop productivity.  With important exceptions, most long-term agricultural

studies, however, are not part of larger networks and operate as place-based studies.  These important studies also remain incompletely inventoried (Richter et al., 2007).

Many of the place-based agricultural studies have made major contributions to the environmental sciences in addition to their intended contributions to agronomy. Perhaps the finest example is Rothamsted's Broadbalk wheat experiment, a field experiment known for its

agronomic data based on 175 years of continuous cultivation.  The Broadbalk wheat experiment may be as valuable for its contributions to the wider environmental sciences as for its contributions to agronomy. Broadbalk publications have been fundamental to quantifying and modeling up to 150 years of changes in: soil fertility, soil carbon sequestration, soil acidification, nitrogen cycling, nitrate and phosphate leaching into groundwaters, adverse effects of industrial

air pollution, microbial community composition, and persistence of potentially toxic compounds (Leigh and Johnston, 1994).  Jenkinson (1991) suggested that Broadbalk's success owed much to the ability of the Broadbalk's managers to periodically modify the themes of research to keep the long-term experiment relevant to societal needs, lessons clearly important to contemporary environmental research networks.


### 4. Contemporary environmental research networks

As geologists debate Earth's transitions over all geological time periods, the time period from the Holocene to Anthropocene Epoch (Waters et al., 2016) is particularly important, as a variety of environmental research networks are quantifying biogeophysical changes in the planet from local to global scales. A foray online can find environmental networks engaged in montoring changes in: lakes (Sier and Monteith, 2016); soil organic carbon (Smith et al., 2002); wind erosion (Webb et al. 2016); agricultural ecosystems (Robertson et al., 2008), to name a few.  There seems to be growing interest in new environmental networks as demonstrated by the recent launch of a mycorrhizal research network in South America (Bueno et al., 2017), and proposals for an ambitious but yet to be funded Long Term Ecological Observatory network in India (Thaker et al., 2015).

Of the variety of environmental research networks, we focus our attention on four: the International Long Term Ecological Research programs (ILTERs), the Critical Zone Exploration Network and Critical Zone Observatory programs (CZEN and CZOs), the Earth or Ecological Observatory Networks (EONs), and FLUXNET (the global network of flux towers that estimate land-atmosphere exchanges of energy, water, and carbon).  While building networks is never easy, each of these is experiencing remarkable success with regards to infrastructure deployment, scientific output, and in training next-generation scientists.  We briefly give an overview of each of these networks and then make suggestions about the scientific and engagement opportunities that the biogeosciences may bring to each.

### 4.1 ILTERs

By the late 20[th] century, the accelerating pace of environmental change has made it incumbent on scientists to get the most out of long-term place-based environmental research sites. In 1984, a paper in *BioScience* by James Callahan was one of several at the time to lay out the case for *networking* long-term ecological research sites (LTERs).  Rallying support for LTER science, what was special about Callahan's paper was that, despite being an NSF program officer, he so sharply criticized NSF's traditional, short-term ecological research programs. Remarkably, Callahan (1984) argued that NSF's short-term ecological research had been counterproductive to the science of ecology, a science that deals with phenomena occurring over decades or centuries and large spatial scales as well.  Today, the USA's robust LTER program, funded from throughout NSF but mainly by NSF's Directorate for Biological Sciences, includes 28 long-term research sites, primarily in North America but also at strategically placed international sites. The LTER research is well known to be question driven,

experimentally designed environmental research and monitoring.  The data collected are meant to answer specific questions and test hypotheses about ecosystem productivity, organic matter recycling, elemental cycling, biological populations, and disturbance. The U.S. LTER sites also function as well-funded focal points for intensive, interdisciplinary, place-based research.

LTER science became international with the formation of the International Long-Term
Ecological Research (ILTER) program in 1993. Ecologists in many nations saw the opportunity and need for international collaboration among long-term ecological research sites to better quantify ecological change across spatial scales.  Today in 40 nations, the independent ILTER Association includes about 800 place-based LTERs, and an impressive array of facilities, scientific expertise, and enormous data legacies with time series that span over a century and
increasingly standardized metadata (Mirtl et al. 2018). The LTERs are located from the Arctic to Antarctica and study forests, prairies, tundra, deserts, cities, agricultural fields, and a variety of estuarine, near-shore coastal, and coastal ocean sites.  All share the common goal to better understand and predict the structure, function, services, and human-altered changes in the Earth's diverse ecosystems.  Forty nations are involved today, and there appears to be good
potential for future growth, as illustrated by the high quality of long-term research ongoing in nations such as Argentina (Contreras et al., 2012), India (Thaker et al., 2015), and Gabon (Braun et al., 2017). There is also growing interest in LTER research throughout the developing world (e.g., Kim et al., 2018).

**4.2  CZEN and CZOs**

In 2001, Earth and ecological scientists in the USA's National Research Council (Ashley, 1998; National Research Council, 2001) defined the concept of Earth's critical zone to be the life-supporting, superficial planetary system extending from the near-surface atmospheric layers that exchange energy, water, particles, and gases with the vegetation and ground layers, down
through the soil to the deepest of bedrock's weathering fronts, extending in space and time the venerable ecosystem concept (Richter and Billings, 2015). The critical zone concept is entirely congruent not only with the ecosystem but with Vernadsky's (1998) biosphere (Figure 3). Critical zone science forces researchers to collaborate on studies of the processes that maintain Earth's life-supporting systems, whether they be expert in the climate, weather, glaciers, snow
and ice, surface or ground water, vegetation, soil, regolith, or the underlying bedrock or sediments (Brantley et al., 2007).  Given that these systems are being altered intensively and extensively by human activities, critical zone science welcomes scientists and scholars who focus on human forcings.

In 2005, the Critical Zone Exploration Network (http://www.czen.org/) was launched by
the Earth science community to stimulate a worldwide community of researchers and educators
who study the structure and processes of the critical zone (Brantley et al., 2006). The CZEN has
helped build an active international community of scientists, many of whom are attracted to the
annual American Geophysical Union (AGU) meetings. A handful of nations or collections of
nations have funded CZO networks, including the USA (Brantley et al., 2017b), France
(Gaillardet et al., submitted), Germany (Zacharias et al., 2011), the European Union (Banwart et
al., 2017), and China (Tahir et al., 2016).

To date, CZO research designs are wide ranging and united by their shared critical zone
concept and that critical zone science must be interdisciplinarity and integrative. Beyond this,
there is no special protocol for how a CZO is to be designed. This has resulted in a wide latitude
in the organization and operation of CZOs. In the USA, nine heavily instrumented CZOs test
place-based hypotheses (Brantley et al., 2017b); but because the nine CZOs span climate,
geologic, and land use gradients (e.g., Chorover et al. 2011), the research teams have each
developed integrated approaches to the study of the dynamic structure and processes of critical
zones within and across observatories. In contrast, the new French CZO program called
OZCAR is a network of networks whose organizational structure differs greatly from that of the
USA's CZOs. The OZCAR program, formally launched in 2015, works on critical zone science
within and across networks of river basins, peatlands, glaciers, reservoirs, aquifers, and
agricultural systems, with all together many hundreds of sites (Gaillardet et al., submitted). And
in China, supported in part by the joint China-UK critical zone science program, long-term
ecological stations are being transformed into CZOs, by adding more geological observations
and opening some of the world's first urban CZOs (Zhu et al., 2017b).

Giardano and Houser (2015) listed 64 CZO projects world-wide; however, additional
CZOs can be added to this list that are extremely important, including Mexico City's wastewater
irrigation CZO (Siebe et al. 2016), a monsoonal CZO in the Western Ghat Mountains of India
(30 June 2018, http://www.czen.org/content/international-czo-working-group), and the new
Ogooué River Basin CZO (Figure 4) in Gabon Central Africa (Braun et al., 2017). What unites
all CZOs and CZO networks is the critical zone concept and questions about how to monitor,
measure, and model the dynamics of critical zone structures and processes as affected by
climate and land use changes.

**4.3 EONs**

The emergence of Earth and Ecological Observatory Networks (EONs) marks a new approach to environmental research networks. Rather than the hypothesis-driven approach of long-term agricultural experiments, ILTERs, and CZOs, the EONs use surveillance-based,

distributed approaches to environmental monitoring and research.  The EONs use surveillance-based, distributed approaches to environmental monitoring and research.  The USA's National Ecological Observatory Network (NEON), Australia's Terrestrial Ecosystem Research Network (TERN), and the Global Earth Observatory System of Systems (GEOSS), are three examples of EONs that use spatially distributed monitoring sites across regional and continental

environmental gradients. The instrumentation has tightly control protocols and large streams of often real-time data are collected, stored, and shared with the wider scientists and managers. The Group of Earth Observations (GEO), a partnership of over 100 governments and nearly 100 organizations, has promoted Biodiversity Observatory Networks as a societally relevant theme in the formation of these EONs.

With EONs, the emphasis is on collecting biologically and ecologically relevant data across wide spatial areas (Walters and Scholes, 2016), data that are made available to the wider research community for analysis. The intent is to identify ecological patterns over time that may not be visible using smaller data streams created from individual research sites. Given the novelty and the growing implementation of EON approaches, Lindenmayer et al. (2018) argue

that there is an "urgent need to find an optimal balance between, and the amount of funding dedicated to surveillance versus question-driven research and monitoring."  They urge an integration of hypothesis- and place-based networks such as the ILTERs and the environmental monitoring networks such as the EONs to take advantage of positive features of both.

**4.4 FLUXNET**

The study of energy, water, and carbon fluxes within and between ecosystems was developed not only by ecologists such as Hutchinson and Lindeman, but also by physical scientists interested in the fluid dynamics and mass and energy exchange at the Earth's surface. Despite an absence of instrumentation, Reynolds (1895) established the fundamental

theoretical framework for the eddy covariance approach to flux measurements, and throughout the 20th c., an international group of physical scientists contributed to the theory and instrumentation of flux measurements (Moncrieff et al., 1997; Baldocchi, 2003). In the late 20th c. these scientists contributed to a growing understanding of two major environmental problems: the effects of large-scale air pollution problems that had spread across Europe and North

America, and the interactions of fossil-fuel driven increases in atmospheric $CO_2$ and the

ecosystem-atmosphere exchanges of carbon, water, and heat. However, not until the collapse of what was then called flux-gradient techniques (Raupach, 1979) and major advancements made in anemometer, gas sensor, and computer technologies could eddy fluxes of energy, water, and carbon be measured. Year-round measurements of ecosystem-atmosphere

exchange were first made in the early 1990s (Wofsy et al. 1993), and by 2000, over 100 flux sites were measuring energy and mass exchanges of ecosystems throughout the world. While ecosystem metabolism was clearly understood by Hutchinson and Lindeman and the Odums (1956, 1968), it took 100 years for physical scientists to assemble the theory and tools to directly measure fluxes of energy, water, and carbon necessary to estimate whole ecosystem

photosynthesis and respiration.

In 2001, the FLUXNET project was established to promote the networking of science and data management of eddy covariance flux towers (Baldocchi et al. 2001), a network that has grown to over 900 historic and ongoing sites, and a network science that has substantially increased our understanding of the dynamics of ecosystems and their interactions with the

atmosphere (Baldocchi, 2014). Today towers operate on six continents and their latitudinal distribution ranges from about 40°S to 75°N. Ecosystems include conifer and broadleaf forests, crops, grasslands, wetlands, and tundra. FLUXNET compiles, archives, and shares flux data via a long-running website (http://fluxnet.fluxdata.org/), and has accumulated enormous data sets pertinent to ecosystem primary production, respiration, evapotranspiration, and sensible

and latent heat fluxes (Chu et al. 2016). While most flux towers have accumulated continuous data for up to five years, a number of towers have accumulated >20 years of data. The network has promoted instrument calibration, post-processing, and reliable gap-filling techniques, and strives to ensure that data among sites are intercomparable. It also supports synthesis, discussion, and communication of ideas and data via its website and workshops. Analyses of

these data sets have facilitated the adaptation of machine-learning (Tramontana et al., 2015) and site-level as well as regional gridded products have helped parameterize and verify biosphere and land surface models (Van den Hoof et al., 2013), and the analysis of satellite remote sensing and global atmospheric measurements (Bonan et al., 2011). The flux tower approach is not without its challenges and we discuss its tendency to focus on aboveground

vegetation and the atmosphere, but evenstill it contributes mightily to the tools and theory of environmental scientists.

**5. Biogeoscience and environmental networks**

This paper asserts the potential for biogeoscience to benefit ongoing environmental
research networks such as those described above. By this we mean that networks can better
keep pace with scientific advances and engage with broader communities by employing an
explicitly biogeosciences approach.  Environmental research networks are major societal
investments, and it is incumbent that science and society get the most from such public outlays
of financial and intellectual capital. The gravity and fast pace of environmental and technological
change mandates that we bring the best of our disciplinary specialties to defining and resolving
environmental problems together, as scientists and scholars. We detail a number of concrete
examples to illustrate how an integrative, biogeoscience approach can enhance network value.

While these networks were founded and grown by remarkably interdisciplinary scientists,
the preponderance of expertise and funding streams have tended to gravitate to different
networks by discipline: ILTERs and EONs  toward ecology and biology, CZOs toward the
geosciences, and FLUXNET toward ecophysiology and micrometeorology.  While our paper's
interest and objective is not to homogenize environmental research networks, we do assert that
biogeoscience presents special opportunities for integrating diverse disciplines in ways that will
benefit the research networks in advancing science and disseminating their science narratives
among scientific communities and the public. We use several examples to illustrate this point.

### 5.1 Biogeoscience and ILTERs

Biogeoscience can potentially enrich ILTERs by bringing advanced geoscience to the
otherwise strong ecological focus of the ILTERs. With notable exceptions, few ILTER sites
characterize well the dynamic structures and biogeoscience processes of subsoils, regoliths,
groundwater, and weathering rock and sedimentary substrata. These are the lower components
of the rooting zone, the water storage and drainage volumes of ecosystems, and environments
that exchange reactive gases produced and consumed by biota and minerals. These are often
heterogenous environments undergoing weathering, the biogeochemical reactions that produce
solutes and new minerals (Schroeder, 2018). Here, we consider research opportunities related
to 1) drilling projects such as is exemplified at the Hubbard Brook ILTER, and 2) the research of
the critical zone ecosystem in the Mezquital Valley of Mexico and 3) the Luquillo ILTER/CZO in
Puerto Rico. Our intent is that these examples may give environmental network scientists ideas
for future studies that draw biological and geological disciplines together.

### 5.1.1  Deep drilling in ILTERs

Coring and drilling campaigns in ILTERs, down through soil, regolith, and underlying bedrock, can open new opportunities to learn how ecosystems function as watersheds and landscapes across space and time. New geophysical techniques are available to quantify and visualize these understudied subsurface environments (eg., St. Clair et al., 2015; Riebe et al., 2016). The Hubbard Brook LTER in North America has had a drilling program with the US Geological Survey that has gathered physical, chemical, and isotopic data of groundwater from wells up to 100 m in depth (LaBaugh et al., 2013). Especially ILTERs with watersheds, streams, and lakes, novel geophysics and geohydrologic approaches can not only characterize subsurface architecture (St. Clair et al., 2015) but can also trace water storage and movement through landscapes (Fan, 2015; Evaristo et al. 2015). Drilling projects can thus help quantify linkages of the atmosphere, terrestrial ecosystem, and aquatic systems, and reveal how ecosystems have functioned and evolved over Earth's history. For ILTERs with interests in dating landforms, drilling campaigns in conjunction with cosmogenic isotopic analyses can provide chronologies of landscape evolution and provide new understandings of erosion and the biogeochemical cycling of nutrients and trace elements (Bierman and Nichols, 2004).

### 5.1.2 How deep is an ecosystem?

That the underlying geological components of watersheds are relevant and integral to the concept of ecosystem was the basis for a few early ILTER studies (Figure 5), but perhaps nowhere more so than in the extremely valuable, underfunded, and decades-long study in Mexico City and the nearby Mezquital Valley. Here the direct connections of land use management and the deep subsurface can serve as motivation for ILTERs and all Earth scientists to deepen their perspective of the ecosystem. The Mezquital Valley has received nearly all of Mexico City's untreated waste water for a century. The valley has a semi-arid climate and is currently irrigated with about 2 m per year of untreated wastewater over nearly 100,000 ha (Siebe et al., 2016). As a consequence, agricultural crops and their human communities have thrived in the valley where today nearly 500,000 people live, entirely dependent on the waste water and its nutrients supplied by the ever-growing Mexico City. For nearly 25 years a team led by Christina Siebe of the University of Mexico has quantified the transformation of the Mezquital Valley as a critical zone ecosystem, making measurements down to 35 m depth, including effects on water, nutrients, trace metals, pathogens and human health (Guédron et al., 2014, Siebe et al., 2016). This CZO also documents the fate of pharmaceuticals and changes in resistance genes during long-term wastewater irrigation (Dalkmann et al., 2012; Jechalke et al., 2015).

### 5.1.3 Co-locating ILTERs and CZOs

The biogeoscience approach to the ecosystem is demonstrated well at the co-located Luquillo ILTER and CZO in Puerto Rico. At Luquillo, ILTER and CZO scientists conduct joint studies of biotic-lithologic controls of surface water chemistry (McDowell et al., 2013); of biogeochemical redox reactions that involve organic matter, redox active metals, and biologically mediated gases (Hall and Silver, 2015, Hall et al., 2016); and of biogeochemical processes that cross wide spatial and temporal scales, i.e., what we are accustomed to calling ecological and geological timescales (e.g. Shanley et al., 2011; Brocard et al., 2015; Dialynas et al., 2018). The research demonstrates how lithologic knickpoints that cross streams and resist downcutting control up-catchment soil formation, plant productivity, and landscape evolution (Wolf et al, 2016). A major contribution to both the biogeosciences as well as to ecosystem and Earth sciences is a revision of the recent geologic history of the island of Puerto Rico (Brocard et al., 2015, 2016), which has major implications for understanding the current distribution of biota on the island as well as understanding the processes that have shaped the landscape over the last 5 million years. The work supports the veracity of the argument made by early LTER scientists such as Callahan (1984) and Swanson and Franklin (1988) that there is profound overlap rather than set boundaries across the Earth's life-supporting space-time continua (Figure 6).

### 5.2 Biogeoscience and CZOs

The biogeosciences can potentially enrich CZOs by bringing novel and advanced biological and ecological research to the otherwise predominant Earth science focus of the CZOs (eg., Chen et al., 2016; Brantley et al., 2017a). The critical zone is called "critical" because all of Earth's diverse life forms depend entirely on the structure and function of the critical zone. Thus, while ongoing CZOs characterize plant and microbial composition and quantify biologically mediated processes, relatively few CZOs characterize: 1) microbial and microbial genetic responses to environmental cues that drive biogeochemistry, 2) the rate and depth to which vegetation may alter the biogeochemistry of the critical zone, 3) depth distributions of biological signals within the critical zone, and 4) dynamics of animal-critical zone interactions.

### 5.2.1  Molecular Biology

The ongoing revolution in molecular biology holds particular promise for better understanding of deep subsurface Earth environments (e.g. 0.5-30 m), which are among the more poorly studied microbiomes on Earth. This revolution affects all environmental research networks, CZOs, ILTERs, and EONs, and FLUXNET, as scientists cope with the burgeoning understanding of molecular biology. Gene surveys of microbial communities in previously underexplored environments, in aquifers, soils, and what Hug et al. (2016) call the "deep subsurface", continue to indicate the existence of enormous numbers of branches in the tree of life, and are prompting fundamental changes in how we understand life's diversity. The new analyses illuminate not only the identity of organisms but their metabolic capacities, with the diversification of bacteria particularly notable (Brown et al., 2015). Nearly all of these on-going findings are not yet represented in biogeochemical models that simulate microbially mediated processes from local to global scales. More systematic measurements of microbial genetic responses to changing land use and climatic conditions can most certainly enhance understanding of critical zone biogeochemistry.

The UK-China Critical Zone program is quantifying flows and transformations of genetic information as an integral, dynamic component of Earth Critical Zone. The resistome of microbial antibiotic resistant genes (ARGs), which has emerged and spread globally in the past 50 years, provides a genetic marker of the Anthropocene. The study of ARGs has the potential to link quantitative genomic information to broader Critical Zone biogeochemical cycles and transformations of mass and energy (Zhu et al., 2017a). In China and other locations, there is major uncertainty about the significance of ARGs in agricultural soils and waters receiving N and P recycled directly from animal and urban human waste water (Zhu et al., 2018).

### 5.2.2 Rapid Vegetation-Critical Zone Transformations

The rate at which vegetation can drive processes in the critical zone ecosystem is not well quantified, but may well be underestimated. In the Pampas of South America, deep soil and groundwater sampling in tree plantations within the grassland matrix demonstrates how vegetation can rapidly transform soil and groundwater chemistry through altered water and nutrient cycling. In these subhumid flat landscapes dominated by herbaceous vegetation, rapidly growing Eucalyptus forests switched the water balance from positive to negative, and led to rapid salinization of soil and groundwater within the root zone (Jobbágy and Jackson, 2007). Remarkably, the Eucalyptus stands salinize soils below the top meter but substantially acidify the soil's surface horizons due to excess cation uptake and sequestration in woody biomass (Jobbágy and Jackson 2003). These decades-long studies and others (Markewitz et al. 1998)

indicate the intensity and rate at which plants can leave unique and persistent chemical imprints on the critical zone ecosystem.


### 5.2.3  Depth of Biotic Signals

While relatively few CZOs have investigated the depth of biotic signals down through the regolith deep into the critical zone, Billings et al. (2018) recently documented that land use history has strong influences on depth of rooting and biogenic agents of soil development, and
that these influences were only partly restored by many decades of forest regeneration.  While root density decreased sharply from 0 to 2 m depth across all soil profiles, below 70 cm root densities averaged 2.1-fold greater under old-growth forests than those in ~70-year old forests regenerating after decades of agricultural cultivation (Billings et al., 20. Differences in rooting were associated with differences in biogeochemical environments in several ways, including
microbial community composition that varied with land use throughout 0 to 5-m soil profiles. Relative abundance of root-associated bacteria was also greater in old-growth forest soils than in regenerating forests. Old-growth forest soils also exhibited a greater fraction of soil organic C as extractable down to 5 m, hinting that more root and microbial exudates were present deep in the subsoil compared to regenerating forests. While both forests had higher $CO_2$ and lower $O_2$
at 3 and 5 m compared with cultivated fields, soil $CO_2$ was higher and $O_2$ lower under old growth hardwoods than under 70-year old regenerating forests (Brecheisen, 2018).  The data suggest that forest conversion to frequently disturbed ecosystems limits deep rooting and biotic generation of downward propagating weathering agents.  Remarkably, some effects of surficial land uses were magnified by soil depth due to positive relations of depth and residence time.


### 5.2.4  Animals and Critical Zones

Animals also can impart important imprints on the belowground ecosystem.  Despite early recognition by Darwin (1882) and Gilbert (1909) that animals shape soils and the surface of the Earth, there is today relatively little study of the fundamental interactions of biology and
soil geomorphology (Paton et al., 1996). Recent work by Winchell et al. (2016) on pocket gophers (*Thomomys talpoides*) in the Boulder Creek CZO in the USA demonstrates the great potential for this field of biogeoscience. Subalpine meadows are habitats for gophers that are small but fierce diggers whose burrows are complex and up to 100 m in length.  The gopher's subterranean habitat provides protection from predators and from winter's cold and provide
access to plant roots for food. The digging and excavating resurfaces the meadows in 50 to 100 years, erosion is accelerated depending on slope, and stone lines are created at about 15 cm

depth. Remarkably, at no time do the gophers enter the nearby forests, so this is fundamentally an animal-vegetation-critical zone interaction. Given that the extent of forest and meadow vegetation at these elevations is affected by wildfires and climate, this bioturbation of the soil
geomorphology ebbs and flows through time.  In spite of some remarkably well-documented studies (e.g. Platt et al. 2016), bioturbation of the upper regolith is a tremendously understudied field of inquiry that could be much more comprehensively studied in all of the environmental research networks.

**5.3  Biogeoscience and EONS**

Nearly all EONs are being created by ecologists and biologists to quantify changes in organisms and the environment over decadal timescales and at regional to global spatial scales. Motivating these networks is a focus on changes in land use and climate, and biotic responses such as species stress and extinctions. The EON data sets that are sometimes compared with
those from networks of weather stations, can be enormous and composed of multiple time series.  Supplementary collections of data are important but the uniqueness of EONs is their common and controlled protocols applied to sites selected to range across environmental gradients.

**5.3.1  EONS and the subsurface system**

The biogeosciences can enrich EONs by bringing advanced geoscience to the strong ecological focus of these networks.  Here, we specifically consider two notable EONs, NEON in the USA and TERN in Australia, both of which have instrumented sites and flux towers arrayed across continental scales collecting data that are conventionally considered to be ecologically
and biologically relevant. We advocate for additional sensors in at least some of these sites to produce datasets that broaden perspectives of environmental change to include the biogeosciences.  We suggest that the scientific impacts of the work to design, construct, and gather initial data at NEON, TERN, and other EONs can be multiplied with a more biogeoscience perspective of environmental change.  The marginal benefits of even selective
placement of sensors deep in the subsurface ecosystem could be large.

Considering the USA's NEON, for which observations are currently made no deeper than 2 m, a design modification strategy might expand this EON's scope via installations of sensors and samplers across the biogeochemical weathering profiles down below the water table and into and through the weathering bedrock. Hydrologic measurements can be
enhanced, geomorphological investigations conducted, and landform evolution modeled using

many novel geoscience tools, including cosmogenic isotopes. We suggest this be accomplished at NEON's and TERN's core sites via proposals from multidisciplinary scientists to guide sensor installation at >2-m, deep borehole sampling, and in applying the latest geophysical, geochemical, and geobiological approaches.

625       Engaging the biogeoscience community in EONs will have reciprocal benefits for the ecological research community. Such engagement will strengthen support for and commitment to EONs among researchers today and in the next generation by providing more early career scientists with opportunities for involvement in these powerful research platforms. EONs, in turn, might well achieve higher quality science as a result of additional intellectual input from

scientists, increased flexibility in their operations, and continued strength in their networked instrumentation platform.

### 5.4  Biogeoscience and FLUXNET

          Eddy flux towers have been versatile in addressing a wide variety of scientific questions

related to climate, ecological gradients, air pollution, rising $CO_2$, and changes in land use and management. Groups of flux towers have estimated how fluxes of energy, water, carbon, and trace gases are affected by plant functional types, length of growing season, drought stress, and disturbances from fire and insect infestations. Flux towers have quantified effects of agricultural management practices, such as fertilization, irrigation, and cultivation; ecological restoration;

deforestation, afforestation, and reforestation; and grazing. Flux tower networks have also provided information on how albedo, temperature, and evaporation vary with climate and ecological dynamics (Baldocchi, 2014). The detailed and continuous records of land-atmosphere exchanges of carbon dioxide have contributed fundamentally to ecosystem and atmospheric sciences, and in particular to understanding hydrologic cycling and ecosystem

metabolism.  For good reason, EONs often have flux towers that are core to their instrumentation, as do many ILTERs and CZOs.

          Here, we consider how flux tower science may advance our biogeoscientific understanding of the ecosystem exchanges of water and trace gases, and ecosystem metabolism.


### 5.4.1  FLUXNET and the water cycle

          Generally, those flux tower sites that measure soil moisture do so at relatively superficial depths. A few tower sites have measured soil moisture throughout the full depth from which water is taken up for evapotranspiration, and partition the origin of transpiration water among

specific soil layers in the root zone.  While few studies have attempted to fully close the above and belowground water budget, several have and these reveal some important results.  Two are reviewed here.

Firstly, soil moisture contents and groundwater dynamics between 7 and 12 m were monitored over four years in an oak savanna with trees dominated by blue oak, *Quercus*

*douglasii* (Miller et al. 2010).  The climate was Mediterranean and semi-arid and the site was in the Sierra Nevada foothills of California.  Soil water storage is small and the rocky A and B horizons only 35 to 60-cm deep over fractured metavolcanic rock. A variety of water storage and flux measurements were collected over four years, including deep groundwater levels, soil moisture contents, sap flows to derive transpiration, and evapotranspiration from eddy

covariance measurements, all aimed at partitioning the amount of water transpired from deep groundwater. Remarkably, groundwater from deeper than 8 to 11 m accounted for about 80% of total evapotranspiration during three months in the dry season. The study concluded that blue oak is not only deeply rooted but probably an obligate phreatophyte, and that groundwater buffers rapid changes in the hydroclimate provided groundwater is not massively

depleted by prolonged drought or by human drawdown.  Secondly, in the modeling study, Thompson et al. (2011) used the variation in the water cycle at 14 flux towers in contrasting ecosystems to compare a "null model" of the hydrologic cycle that coupled the Penman-Monteith equation for evapotranspiration with changes in soil moisture, and explored deviations between the null model and observations of water fluxes from the eddy covariance

measurements. While the null model reproduced evapotranspiration reasonably well in arid, shallow rooted ecosystems, it overestimated effects of water-storage limitation and could not reproduce seasonal variations in evapotranspiration in more humid and more deeply rooted ecosystems. Accounting for root access to deep soil moisture including from groundwater greatly improved prediction of evapotranspiration in the more humid ecosystems across multiple

time scales.  Both research studies (Miller et al., 2010; Thompson et al., 2011) not only demonstrate the value of eddy flux approaches for ecosystem analysis, but they also remind us that ecosystems extend deep belowground and that hydrologic characterization of the deep subsurface may often be needed to close hydrologic budgets.  Remotely sensing soil water has yet to demonstrate an ability to quantify water throughout the root zone.


**5.4.2 FLUXNET and trace gases**

Continued development of sensor technologies has enabled scientists to measure net ecosystem fluxes of many trace gases that were below detection limit in the past. A good

example is methane, a gas whose sensors have been able to estimate fluxes for over a decade,
and a gas long recognized to be an important part of the carbon cycle in peatlands (Gorham, 1991) and one particularly influential to the greenhouse gas forcing of the climate. One of the first studies was that of Rinne et al. (2007) in a boreal wetland fen, who made the significant observation that about 20% of the annual $CO_2$ assimilated by plants was emitted as $CH_4$. Petrescu et al. (2015) used data from a network of wetland flux towers to estimate $CO_2$ and $CH_4$
fluxes concluded that net radiative forcings from the two gases were much higher in wetlands converted to human uses.

Sensor technologies continue to advance rapidly. Tunable spectrometers *simultaneously* measure fluxes of many hundreds of volatile organic compounds (Park et al. 2013), tracers of vegetation and microbial reactions such as carbonyl sulfide and nitrous oxide, respectively, as
well as fluxes of stable isotopes (Griffis, 2013). The prospects for future applications of eddy covariance techniques are most certainly bright.

### 5.4.3 FLUXNET and the metabolism of terrestrial ecosystems

One of the most significant outcomes of eddy flux measurements has been
measurements of carbon fluxes in studies of how ecosystem metabolism is affected by disturbance and recovery.  In a variety of ecosystems, measurements of carbon fluxes between the ecosystem and the atmosphere have documented how clearcutting, thinning, grazing, forest regrowth, woody encroachment, and wildfire and prescribed burning alter ecosystem exchange of carbon (e.g., Law et al., 2003; Amiro et al., 2010). In general, a $CO_2$ pulse from respiration is
observed in the years after disturbance, but usually ecosystem photosynthesis matches and exceeds respiration within a decade, depending on the particulars of the ecosystem and disturbance. Maximum carbon uptake often occurs throughout the remainder of the first century of regrowth, but it is highly significant and even surprising that nearly all old-growth forests with flux measurements are carbon sinks (e.g., Knohl, et al., 2003). Although future studies will need
to more precisely estimate carbon fluxes over the life of forests under different management and disturbance regimes (Harmon et al., 1990), the likelihood that old-growth forests are typically carbon sinks is a major contribution to discussions over Odum's (1969) concept that post-disturbance succession leads to old growth forests with photosynthesis in balance with respiration (Christensen, 2014). Flux tower approaches to ecosystem-atmosphere exchange
have much to contribute to ecosystem and critical zone science, particularly when accompanied by characterization of the subsurface that defines the lower boundary conditions for ecosystem fluxes of energy, water, carbon, and other gas-phase elements and compounds.

### 6. Conclusions

725 Despite the radical interdisciplinarity of many of the founders of the biological and geological sciences, for over a century these two disciplines have developed with relatively little interaction. More often in parallel than together, biologists and geologists have studied the Earth's diverse landscapes and ecosystems; the circulation of water, energy, gases, and chemical elements; and the temporal and spatial dynamics of the planet's living and non-living

730 systems.

 In the best of all worlds, research agencies will expand open-ended requests for proposals to encourage creativity and excellence from teams of ecologists and Earth scientists, specifically within environmental research networks. Both site- and network-based research proposals might be requested to advance the science and engagement with environmental

735 research networks. Proposals for research should be open to post-doctoral fellows and early career scientists, with provision for small projects and travel grants to develop wider participation across the biology, ecology, and geoscience communities.

 Given that the ILTERs and EONs have grown supported largely by the biological sciences, CZOs by the geosciences, and FLUXNET by the micrometeorology and

740 ecophysiology communities, we argue that more explicit biogeoscience activities within these research networks can create opportunities to meet G.E. Hutchinson's challenge (Slobodkin, 1993): to "confront all of the processes that maintain or change ecological systems, whether these processes were (sic) biological, physical or geological."


**Acknowledgments**

*This manuscript is dedicated the late Dr. Henry Gholz, who gave and received much joy in his championing of ecosystem science* (30 June 2018; https://www.nceas.ucsb.edu/news/remembering-henry-gholz).  The manuscript was inspired by

LTER-CZO meetings at the 2015 LTER All-Scientists Meeting in Estes Park, Colorado from August 30 to September 2.  An invitation by the LTER Network had been extended to Dr. Tim White of the CZO Program National Office and two CZO Principal Investigators (McDowell and Richter).  Many co-authors work at environmental research network sites and participated in discussions of the Work Group on CZO-LTER Collaboration (30 June 2018;

http://asm2015.lternet.edu/working-groups/critical-zone-observatorieslong-term-ecological-research-network-collaboration). The lead author thanks Will Cook for manuscript review, Duke University, and the National Science Foundation (NSF) for funding through the Biological Sciences Directorate and through the Geosciences Directorate's Division of Earth Sciences Critical Zone Observatory program (EAR-1331846).


**Author contributions**

Richter prepared the manuscript with contributions from all co-authors.  Most coauthors work at environmental network research sites and many participated in the 2015 LTER All-Scientists

Meeting in Estes Park, Colorado (proposed by Tim White), in 2017 discussions of a Work Group on CZO-LTER Collaboration, and in 2016 and 2017 International CZO discussions at AGU Annual Meetings.

**Author conflicts**

The authors declare that they have no conflicts of interest.

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

**Figure captions**

Figure 1.  The great biogeochemist Vladimir Vernadsky gave special attention to solar radiation driving global photosynthesis and subsequently the global biological, geological, and human responses that follow from this remarkable transfer of "cosmic energy". The false-color composite image displays ocean chlorophyll *a* concentrations from dark blue at ~0.05 mg/m$^3$ to green at ~1 mg/m$^3$ to red at >30 mg/m$^3$, and land normalized difference vegetation index from a minimum as brown to a maximum as dark green to blue. (The image of global photoautotroph abundance is integrated over 20 years from 1997 to 2016, and is provided by the SeaWiFS Project, NASA/Goddard Space Flight Center).

Figure 2. Throughout the 20$^{th}$ century, precipitation and streamflow were measured in watersheds, for example a) at Coweeta Hydrologic Laboratory in western North Carolina where in Watershed 17, catchment hydrologic response to deforestation was quantified (Hursh et al. 1942).  Watershed-ecosystem studies continue to quantify hydrologic responses to land uses but with greatly advanced instrumentation as is shown in b) the Stringer Creek Watershed in Montana (Hood et al. 2006), in which within watershed processes such as water storage, interflow, groundwater recharge, gas exchanges, nutrient flows, evapotranspiration, and other processes are estimated with advanced sensor technologies, often in real time, supported by LiDAR-generated digital elevation models, and eddy covariance fluxes of energy, water, and carbon.

Figure 3. Core conceptual models of a) ecologists' ecosystem (Lindenmayer and Likens 2009, after Bormann and Likens 1967), b) Earth scientists' critical zone (courtesy of Southern Sierras Critical Zone Observatory), and c) an EON design (courtesy of USA's National Ecological Observatory Network), complete with eddy covariance flux tower.  The congruence of the ecosystem and critical zone concepts that guide these networks motivates biogeoscientists to consider how to get the most from these long-term research investments.

Figure 4. Hydrologic stations are currently being installed and upgraded in the vast and understudied central African landscapes. The international French critical zone program, OZCAR, is expanding biogeoscience infrastructure in the Congo River basin as well as in smaller river basins such as Gabon's Ogooué River. The image is of Gabonese savannas and forests in Lopé National Park (Photograph, J.J. Braun).

Figure 5.  An example of geohydrological, geochemical, and geohistorical control over lake water chemistry, biology, and ecology. The diagram (slightly redrawn here) was used by Magnuson (1990) as justification for expanding space and time scales for ecological research. The figure illustrates the relatively slowly operating hydrogeologic processes of groundwater flow through contrasting glacial tills. Dilute water from Crystal Lake (~10 $\mu$mol/L HCO$_3^-$) arrives at Big Muskellunge Lake with greatly elevated alkalinity (~350 $\mu$mol/L HCO$_3^-$) due to flow paths of groundwater that extend tens of meters deep through carbonate-rich substrata. The example illustrates the overlapping of biological and geological sciences in space and time.

Figure 6. A space-time diagram of the sort used to justify LTER's emphasis on extending ecosystem research into longer times cales and larger spatial scales (modified from Wu 1999). Spatial-temporal scales of interest in ecology and Earth sciences overlap from small to large and from instantaneous to many millions of years.

Figure 1.

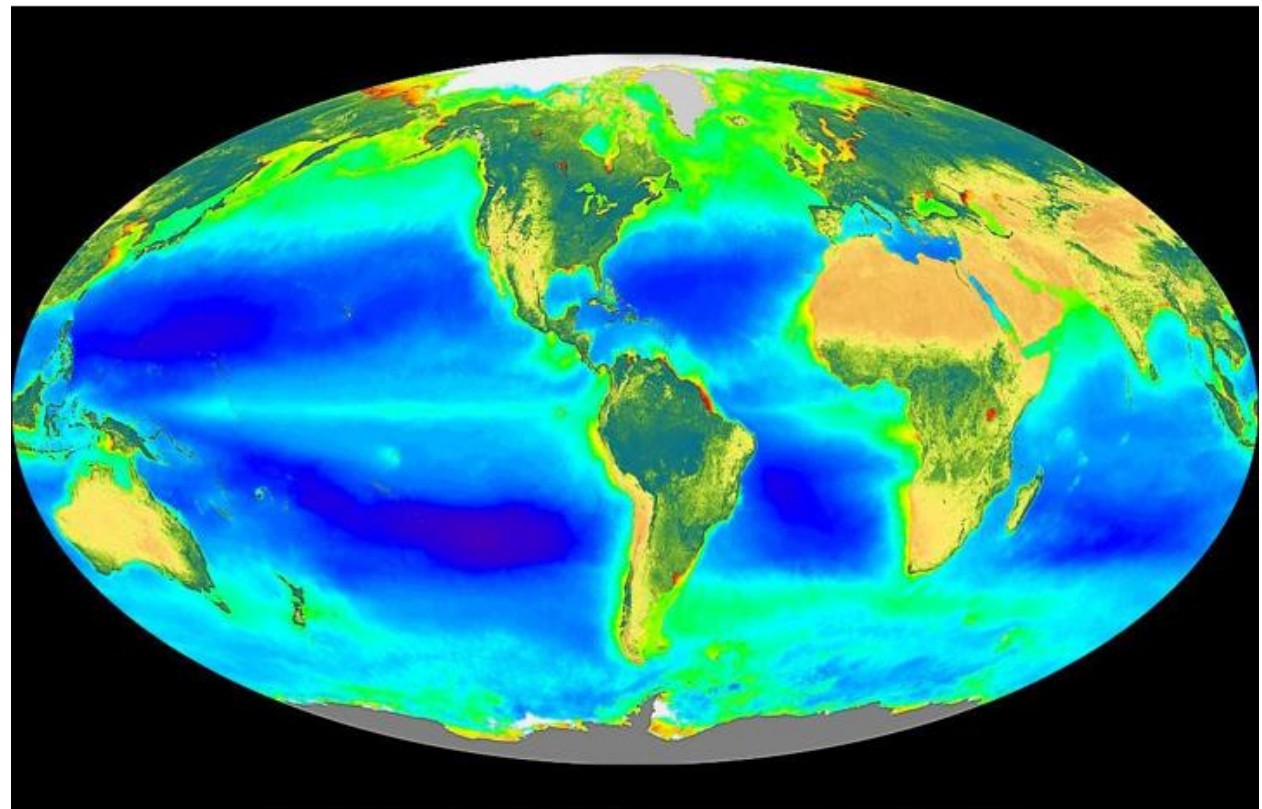

Figure 1.  The great biogeochemist Vladimir Vernadsky gave special attention to solar radiation driving global photosynthesis and subsequently the global biological, geological, and human responses that follow from this remarkable transfer of "cosmic energy". The false-color composite image displays ocean chlorophyll $a$ concentrations from dark blue at ~0.05 mg/m$^3$ to green at ~1 mg/m$^3$ to red at >30 mg/m$^3$, and land normalized difference vegetation index from a minimum as brown to a maximum as dark green to blue. (The image of global photoautotroph abundance is integrated over 20 years from 1997 to 2016, and is provided by the SeaWiFS Project, NASA/Goddard Space Flight Center).

Figure 2.

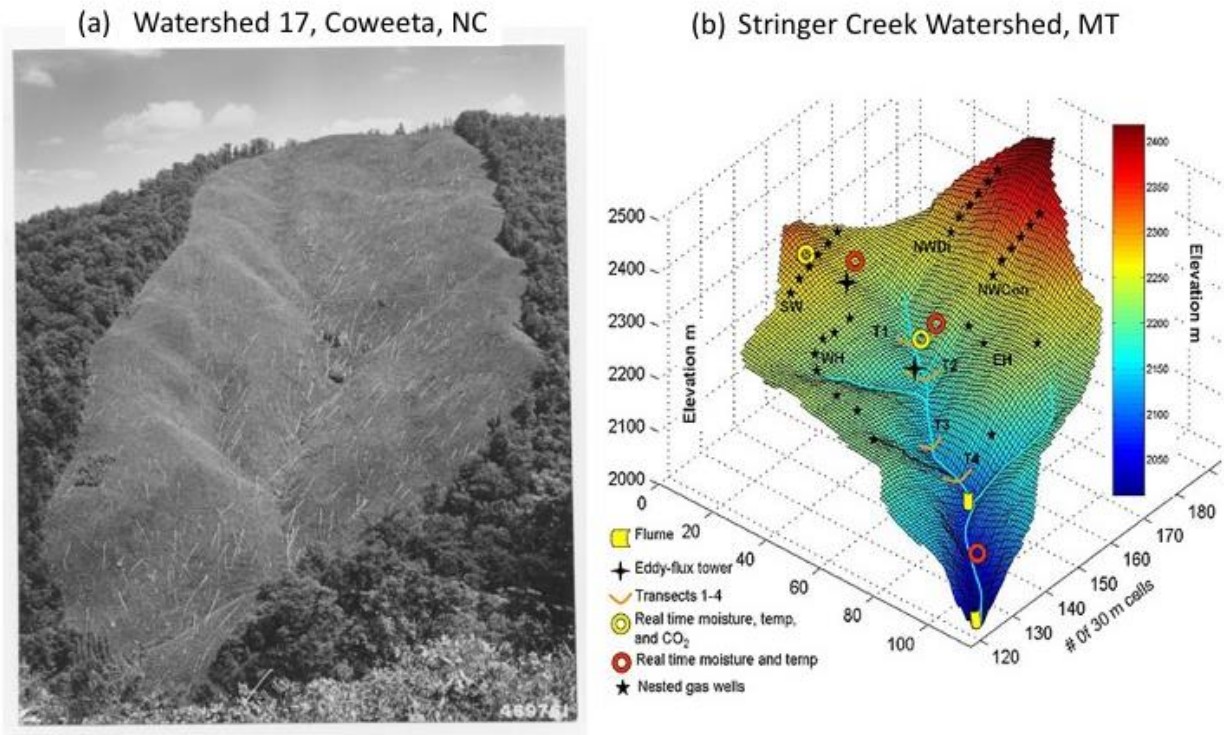


Figure 2. Throughout the 20th century, precipitation and streamflow were measured in watersheds, for example a) at Coweeta Hydrologic Laboratory in western North Carolina where in Watershed 17, catchment hydrologic response to deforestation was quantified (Hursh et al.
1942). Watershed-ecosystem studies continue to quantify hydrologic responses to land uses but with greatly advanced instrumentation as is shown in b) the Stringer Creek Watershed in Montana (Hood et al. 2006), in which within watershed processes such as water storage, interflow, groundwater recharge, gas exchanges, nutrient flows, evapotranspiration, and other processes are estimated with advanced sensor technologies, often in real time, supported by
LiDAR-generated digital elevation models, and eddy covariance fluxes of energy, water, and carbon.

Figure 3.

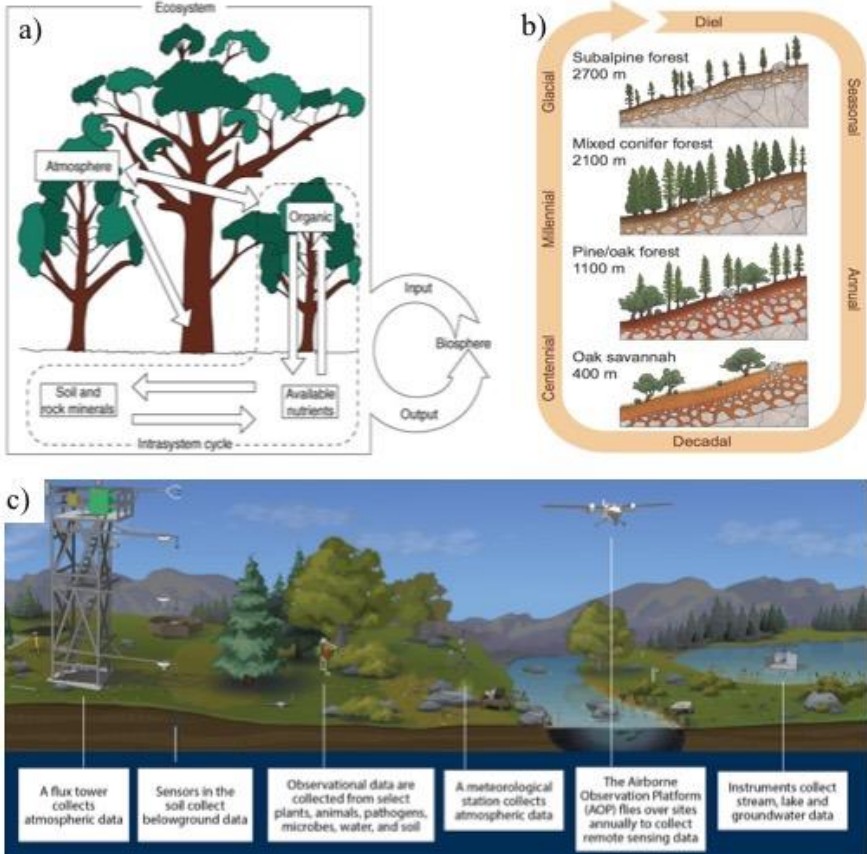

Figure 3. Core conceptual models of a) ecologists' ecosystem (Lindenmayer and Likens 2009, after Bormann and Likens 1967), b) Earth scientists' critical zone (courtesy of Southern Sierras Critical Zone Observatory), and c) an EON design (courtesy of USA's National Ecological Observatory Network), complete with eddy covariance flux tower. The congruence of the ecosystem and critical zone concepts that guide these networks motivates biogeoscientists to consider how to get the most from these long-term research investments.


Figure 4.


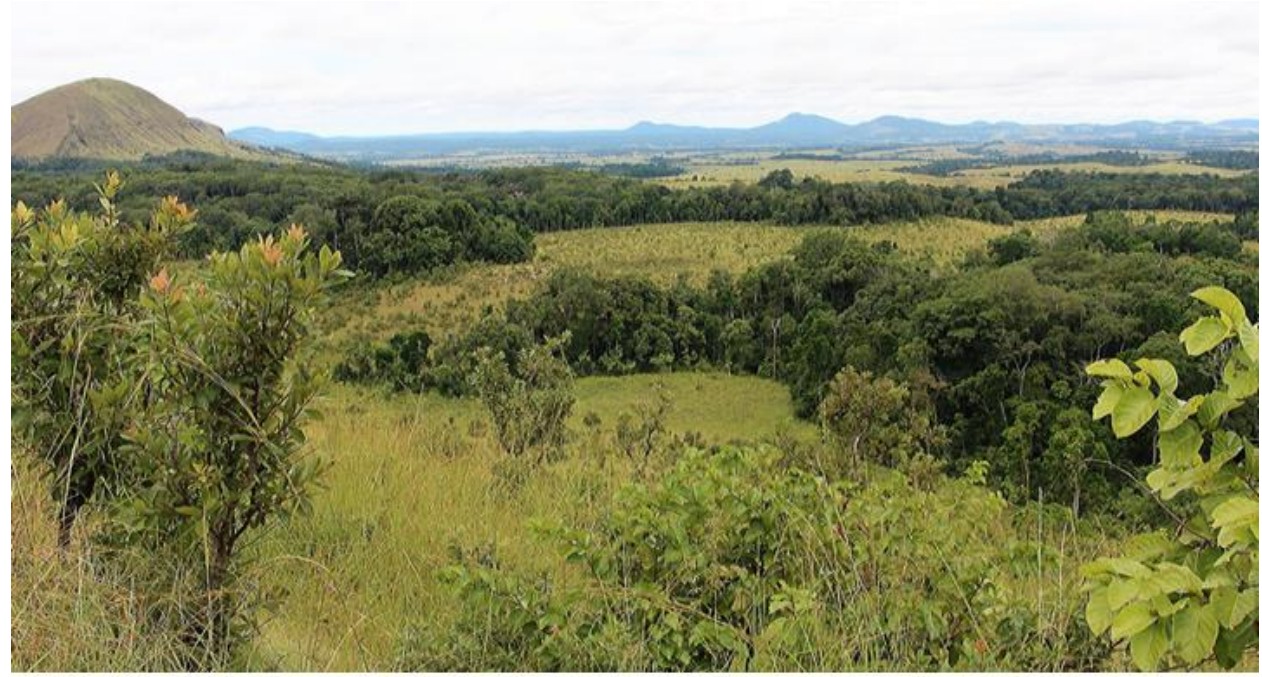

Figure 4. Hydrologic stations are currently being installed and upgraded in the vast and understudied central African landscapes. The international French critical zone program, OZCAR, is expanding biogeoscience infrastructure in the Congo River basin as well as in smaller river basins such as Gabon's Ogooué River. The image is of Gabonese savannas and forests in Lopé National Park (Photograph, J.J. Braun).

Figure 5.

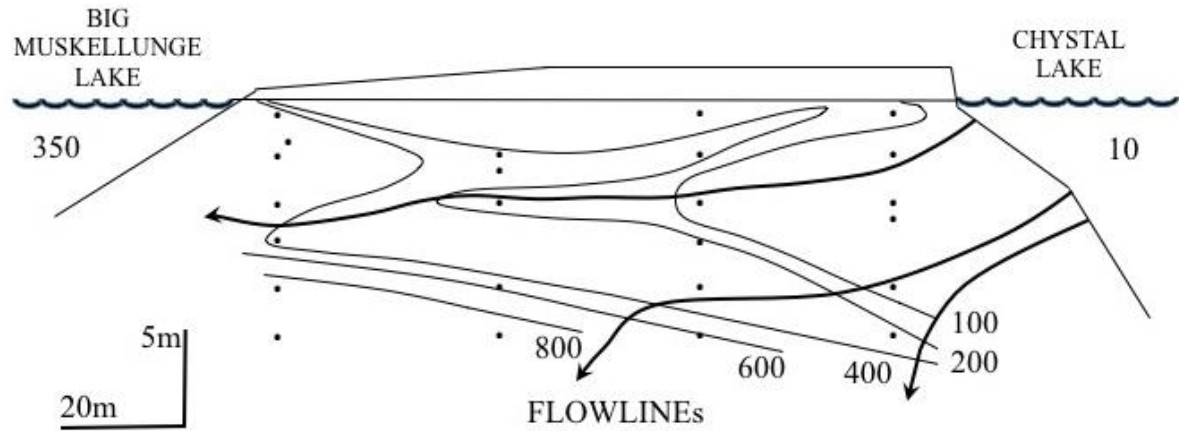

ALKALINITY PROFILES (μmol L⁻¹)

Figure 5.  An example of geohydrological, geochemical, and geohistorical control over lake
water chemistry, biology, and ecology. The diagram (slightly redrawn here) was used by
Magnuson (1990) as justification for expanding space and time scales for ecological research.
The figure illustrates the relatively slowly operating hydrogeologic processes of groundwater
flow through contrasting glacial tills. Dilute water from Crystal Lake (~10 $\mu$mol/L $HCO_3^-$) arrives
at Big Muskellunge Lake with greatly elevated alkalinity (~350 $\mu$mol/L $HCO_3^-$) due to flow paths
of groundwater that extend tens of meters deep through carbonate-rich substrata. The example
illustrates the overlapping of biological and geological sciences in space and time.


Figure 6.

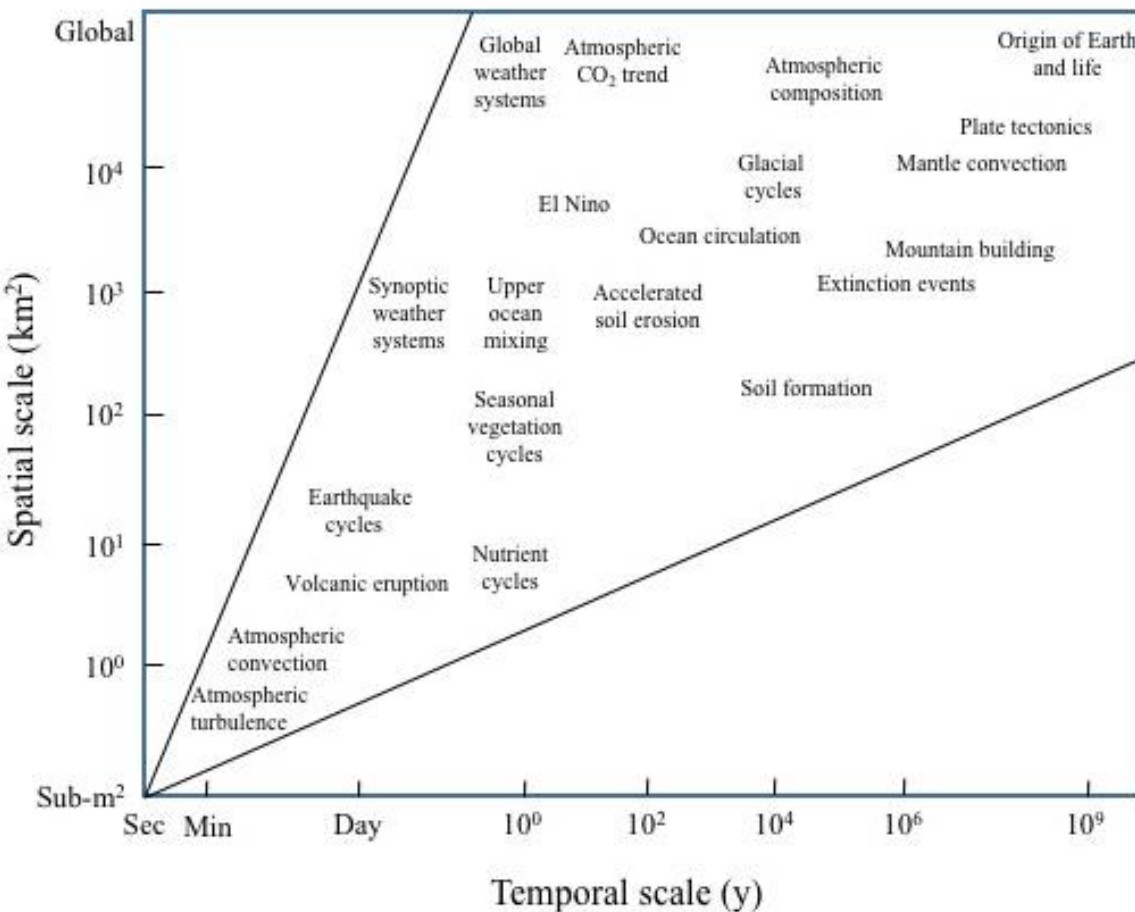


Figure 6. A space-time diagram of the sort used to justify LTER's emphasis on extending
ecosystem research into longer times cales and larger spatial scales (modified from Wu 1999).
Spatial-temporal scales of interest in ecology and Earth sciences overlap from small to large
and from instantaneous to many millions of years.