# Peer review of "Strengthening the biogeosciences in environmental research networks"

_Biogeosciences, 2018_

## Referee Comment (RC1) · H W Loescher (Referee) · 8 Mar 2018

The authors present a very important manuscript advocating better integration of Observatories (EONs) and networks (LTER and CZO) through the biogeosciences. I strongly agree with the authors that this advocacy is needed. The topic of this manuscript is timely and pertinent. I greatly appreciate all the hard work that has gone into this crafting this manuscript.

I am very familiar with most all the co-authors, the observatories and networks, the scientific rationale(s), and working in these organizations. I am also quite familiar with the subject matter. I feel I have broad knowledge of this area of research and development, and feel this is a fair and honest review. I feel that if a reviewer has constructive

and collegial comments, they should not hide behind anonymity. Hence, I do not wish to remain anonymous, and please consider this review signed, Hank Loescher. At this time, I recommend a major revision.

This is a great opportunity to advance Earth system science, but with all due respect to my esteemed colleagues, I think you have missed the point:

First and most importantly, what is being advocated is as much cultural issue in conducting science as anything. I have seen the direct effects of different scientific cultures in numerous, recent meetings with members of the EONS, LTER, CZO and other networks present. In that the authors are well respected members of the ecological community, there is a large responsibility to communicate your message correctly and with an awareness of the cultural and political sensitivities. Advocating change will be best served by communicating your ideas in a way that can be heard by all the respective user communities. As it is written, I find obvious biases in how each of the research structures are described, and this does a disservice to the goals of the paper and to the future user communities. Authors hold onto an old paradigm of how science is best performed (e.g., no faults to the LTER approach), clearly contain their own bias, and do not present a meaningful path forward. There are a lot of misnomers and imprecise statements that also show this bias, and if published as is, would propagate these biases to the user communities. The issues at hand are not about perpetuating the LTER paradigms in the use of 'hypotheses questions' moving forward, but embracing (and owning as a community) new institutionalized approaches (networks and observatories) to challenge our current approaches (get out of our current boxes) and move the science forward with new tools. I do not see that the authors take ownership if this problem. Second, the authors do not clearly and objectively describe the respective strengths, weakness and complementarity of all these research organizations. I strongly recommend an unvarnished assessment of the attributes and approaches of the respective organizations. The manuscript would be better served if the authors provide specific science themes together with the approaches that can integrate the

data and advance our understanding across disciplines, processes, time, and space.

Other key issues EONs (NEON) and networks (CZO/LTER) are BOTH question based. But how they are applied (EONs to inform requirements and top down designs) and Networks (bottom up tradition hypothesis testing) differ. Both utilize the same suite of questions, the nuance is HOW they are implemented. And I fully agree that there has to be a structure in place to revisit, revise, and update the EON capabilities against the rubric of frontier science questions.

The text often jumps from idea to idea, from concept to concept without really discussing the issues or core mechanisms to really bridge disciplines and integrate concepts. Narrative structure needs more synthesis-style of writing.

If you look at the original planning documents for LTER, they look a heck of a lot more similar to an EON than what their organizational structure and function are today. Why is that? The change in organizational approach of networks (LTER) over time is natural in its evolution/development. Acknowledging this and advocating for change in the context of biogeosciences among networks is very natural–and messy way that we do science. Maybe state as much.

Authors do not really take ownership of the issues at hand, or the process of integrative change in EON structures. Rather, they pointing out the shortcomings and advocate the same old, the same old. The only difference in this manuscript is that biogeosciences is being broadly advocated as the integrative theme without any real specifics in how to do this.

specific (nitpicky) comments L112, what 'new systems analyses'?, best to define it a tad more. L122, what 'critical zone scientists'?, best to define it a tad more. L129, the acronym of 'ILTER' is not really defined L133, doesn't 'EOS' stand for something? L135, what 'is a time scale in LTER'?, what is a 'time-scale' in a network? do y'all mean 'site-based LTER science'? or something like that? L141, what 'NEON'?, best to define it a tad more. Peters et al. 2014 provides a working description of 'EONs'

and other ecological infrastructures. L143, what about temporal scales?, re. designed to scale in both time and space... L144, I think the statement of 'ecological conditions and biodiversity' falls at tad short. (i) does not embody the philosophical approach of cause and effect, and (ii) in the case of NEON there are 7 grand challenge areas that were adopted by the NAS 2001, 2003 reports. Moreover, many would argue that no EON can estimate 'biodiversity' very well. The specific approach and rationale for biodiversity observations have to be taken into account, and towards what end? Moreover, biogeochemistry (that y'all are arguing for) is also being measured in NEON, though maybe not necessarily measuring what individual investigators want, or to a desired fidelity. The EON design criteria includes; (i) to be applicable to a broad user community, (ii) to be considered data product/approach broadly accepted by the user community, (iii) data product/approach not be considered experimental, and (iv) under pragmatic and fiscal constraints. All EONs have the same design issue. Best to highlight that there are other EONS internationally (just like that noted for iLTER and iCZOs) L146, 'question-driven' implies that 'EONs' are not question driven. This is not true, and does a disservice to the community and emergent culture of integrated research infrastructures. See note above. L156, while it is nice to see the whole book referenced, (re. Chabbi and Loescher 2017), the point that y'all made was in the chapter; Loescher, H. W., E. Kelly, and R. Lea, 2017 National Ecological Observatory Network: Beginnings, Programmatic and Scientific Challenges, and Ecological Forecasting. In: Terrestrial Ecosystem Research Infrastructures: Challenges and Opportunities. Eds. A. Chabbi, H.W. Loescher. CRC Press, Taylor Francis Group, Boca Raton, FL, pp. 27-48. ISBN 9781498751315. L165, consider '...and Earth scientists, alike...' L167, what does 'best of the biogeosciences' mean? best to be a tad more concise in the writing and a little less arm-wavy. L170 the comment '...to work across these networks to help solve pressing environmental problems and puzzles.' confuscates the difference between the need to advance basic science and understanding with the need to demonstrate societal benefit, economic value and an applied approach. Which is it? Best to be a tad more concise. L170, the voice and tone... reads tad self-serving. The idea to in-

tegrate biogeosciences among several science disciplines has been around for a long time. Authors are correct to point out that reductionism plays a part (historically), but also note that a resistance to change current approaches is much more of an issue today. It seems as though there are a lot of issues are raised without fully embracing a synthetic statement or path forward... (i) basic science, developing a are understanding, and discovery, (ii) applied science toward decision makers (natural resource management), and (iii) policy driven science, all of which have different implication of how science is being done in the context of a network or infrastructure. L181, Schimel, D., M. Keller, S. Berukoff, R. Kao, H. W. Loescher, H. Powell, T. Kampe, D. Moore, and W. Gram, 2011. NEON Science Strategy; Enabling continental-scale ecological forecasting. Pub. NEON Inc., Boulder CO. pp 55. [webpage citation] is a more appropriate reference. L227, why are 'field experiments',Âăimportant? What is the philosophical context that becomes important in your narrative? Someone can say, so what, LTER has experiments?, towards what end? L227 while LTER research embraces different themes, PI based research is question/hypotheses driven, which also can be seen as a limitation, because of its lack of integration among other science that is being done at and among sites regardless of its utility to public policy. See comment above. L233, EON's in operations are not 'projects', particularly in the eyes of NSF. They are Large Facilities, or operational Research Infrastructures. Best to change the text to reflect this. L240, it is not a have '30-y vision', it has an NSB approved operational timelime of 30-y. This is a very different thing. L235, I strongly disagree with the statement 'EONs are not question-based or hypothesis-testing projects', and it does a large disservice to the user communities. They have been informed by grand challenge questions (from NAS in the case of NEON), and investigator based hypotheses. HOW they use them in the design is different. And I fully agree that there has to be a structure in place to revisit, revise, and update the EON capabilities against the rubric of frontier science questions. See comment above. L237, what is a 'highly controlled measurement'?, not a very concise statement. Do y'all mean measured in the same way across all the sites, with the same sources and magnitude of uncertainty, rigorously QA/QC'd, to assure robust cross site analyses? L238, L249, 'intended' sounds a bit arbitrary and argumentative, I would suggest to use 'designed'. L241, not really a network, NSF prefers 'Facility', 'Infrastructure' or Observatory. See comment above. L247, L250-251 it is definitely NOT 'NEON's mission is to analyze and forecast impacts of. . .'. NEON is charged to provide the data to enable an ecological forecasting. NEON is NOT preforming any of the data analyses or forecast ecological processes– that is for the community to do. I am quite surprised that this narrative was crafted this way, given that some of the co-authors are intimately aware of this point. L253, what does '. . .tightly controlled. . .' mean?, all NEON, TERN, SAEON data is open access. . . please be concise in your meaning here. L255, 'short time scales'? Not quite as concise and embracing a narrative that you could use. If you are discussing these data with an ecosystem scientists, they would potentially think decal scale data is very long, if chatting with micrometeorologists, they would think decadal scales would be infinite! I suggest to qualify this statement as something like . . . 'short time scales when compared to geological timescales that CZO community is accustomed to.', or something towards that effect. L257-60, NEON's current design does little to accommodate change. I do not disagree. But the verbiage is adversarial, rather than engaging. L276, 'NSF' is not defined L278-287, awkward sentences, suggest re-crafting it. L290, verb missing?, '. . .pertinent to [understand?] critical zone structure and function.' I do not think a study itself is pertinent to the CZO structure. L291-298, awkward sentence, suggest re-crafting it. L298-305 seems like a laundry list without any real syntheses of why these are important. Moreover, the paragraph begins w/ US CZO and then China, Mexico, France and India are mentioned. Best introduce there is a iCZO network analogous to iLTER. L310-313, redundant, re-write or remove L314, I disagree. A meteorologist is concerned at the synoptic or orographic time scales. Some CZO sites measure the turbulent exchange (much shorter timescales), hence it would be more concise to state 'micrometeorologist'. L316, are they really 'young', or early career? L320, interests? Unclear. L323, 'informational and physical' = good! L323, last clause '. . .and, expertise across LTERs, . . .' seems out of place and a vestige from

edits. L324-333, seems arm wavy, what is different here than already being done (to play devil's advocate) what is really new here? What is the nitty gritty here? Just saying we need it, is not different from what current scientists do... L340, what happened to atmospheric? L390, 'full bloom' is jargon and very odd choice of wording. Suggest crafting the test with more approachable narrative structure. L392-L395, while there are interesting points here, the text fails to synthesis the core integral concepts that are needed to advance our science. Stating that instantaneous to millennial timescales are addressed, but fails to discuss how this is done and to what end. What is the nitty gritty here? L399, what 'benefits'?, and L402, what 'great opportunities'? just stating so, does not make it so. L418-420, How are these ideas being integrated?, merely stating so does not make reality. L431, Advocating a call to action to 'research agencies' is parochial at best. Suggest figuring out a different way to articulate this. Rarely does such a statement effect change in the programmatic activities or funding opportunities of an agency. L465, no Acknowledgements? L469, seems like a verb is missing. L592, is Josh Schimel, not O Schimel.

---

## Short Comment (SC1) · 19 Mar 2018

For the upcoming EGU General Assembly I have to prepare myself on Science Integrity being kindly asked by young scientist to speak on that topic. As thsi topic is very challenging for me, I am carefully scanning literature on topics on "reuniting" Bioscience and Geoscience. Additionally, I am highly interested if such an open discussion process before "real" publication adds to improve Science Integrity. I think it has some potential, but only if outsiders add to the discussion and that is what I do now.

I was delighted having found this beginning discussion which has the potential to serve Science Integrity. I also applause Hank Loescher to identify himself, particularly, as it is a quite fundamental critique he puts forward. I do agree that the authors could be

more precise or more pushy in presenting a path forward (discussing less the present state) – it should be sharpened on what could be new. I definitely would like to have more advice on how to move forward bringing bio and geo back together. However, Hank Loescher himself keeps imprecise in his very general critique – e.g. "There are a lot of misnomers and imprecise statements that also show this bias" - which bias is not clearly stated to my opinion, at least, as an author, I would not know how to improve my work now. So I would like to encourage Hank Loescher to name the proposed biases in greater detail for the sake of Science Integrity. As an outsider, it appears to me more like criticizing with low evidence. I am sure this is not what he intended to do. For me it is more serious than other cases since Hank Loescher is a representative of Eons – one of the three program discussed in the paper – which obviously was not well described to him. To ensure his objectivity, I think Hank Loescher would be well-advised to be more precise on what he called "obvious" biases. There are not obvious to me, but may the authors know better.

Please do not get me wrong Hank Yours Hermann

---

## Referee Comment (RC2) · J. Beringer (Referee) · 21 Mar 2018

Globally, research networks are key to supporting regional and global science. They have not developed equally across the globe and even the progression of networks within a country can be messy. The authors have done a good job at describing the US based networks and the manuscript is generally well written. Therefore, the authors have provided a great platform for discussion and for this reason I think it is important. I am not from one of those networks and I acknowledge that my comments are partly opinion too but hope that it adds to a useful discussion. I will keep my comments general at this time as I think the paper needs major revision.

My first impression is that this is very American centric with the author list and examples

of networks used in the paper. Bringing biogeosciences to LTERs, EONs, and CZOs is necessarily an American focus due to the specific infrastructure programs but should be a global one. Is there a better framework here? Or at least better nomenclature? If, as they propose, we are "interested in addressing questions that motivate the worlds research networks" then this requires a global effort and integration across boundaries is necessary. In addition, submission to Biogeosciences is to a global audience and the paper should be relevant. To stay in Biogeosciences I would recommend that the paper should expand globally to be a real interest and engage our global community rather than assuming that the rest of the world is the same as the USA.

The objective of the paper was to "motivate more collaborators to bring the best of the biogeosciences to the LTERs, EONs, and CZOs". In my opinion this is a rather odd objective to have for the paper and I don't think it achieved it. The paper suffers from not having a clear problem statement, so it is never clear what the discussion is trying to achieve and so in the end it achieves nothing. The scope and problem need to be well developed with some specific outcomes in mind. For example, what are the pressing problems and puzzles of the world that you allude to and how can they be addressed by integrating different measurement capabilities or networks. What the important global questions that you refer to in the paper? It would be useful to determine these perhaps as an integrating framework – find synergies first.

There is also no definition of what biogeosciences is and what a is a 'biogeoscience approach'? Are we talking about Earth system Science or Critical Zone Science?

I agree that these networks are underutilised. Many networks globally struggle with minimal (or no) funding to keep the lights on and funds do not support scientific research. This is a problem particularly for investigators who spend a lot of time running things. Perhaps you could discuss this? The authors ague that "In fact, the core concepts that motivate these networks' operation clearly and substantially overlap (Figure 4), that is, ecology's ecosystem is entirely congruent with Earth science's critical zone". I find this is a cursory observation and the figure simply shows different schematics for

each network and does nothing to synthesis the information. There needs much more development of these ideas here. This would be more beneficial if it were a systematic analysis of the operational and conceptual frameworks associated with each. Maybe a table or synthetic diagram. Same with the differences, make a systematic review of the difference and/or gaps. What are the challenges they face?

"EONs are surveillance facilities" I don't truly understand the distinctions you are trying to make here. All facilities are surveillance really.

At the end of the day the networks (or what they provide) are just tools. They are constantly evolving and being reinvented to attract and maintain funding. Really what we want is a scientific framework that is capable of answering these questions BUT then utilises tools in its toolkit. These would include surface observational capability (LTER, NEON, CZO), atmospheric tools, remote sensing and modelling. Shouldn't we also include the humanities? If that is the case don't we already have this is Earth System Science (relevant programs are iLEAPS, GLP, Future Earth). How do the networks help address the big societal questions that they are posing? Are we reinventing the wheel in biogeosciences because we already have ESS? This needs discussion and thought.

Also the observational facilities you mention are very specific in the US and are quite different elsewhere in the world so there is a huge need to think about this in terms of CAPABILTIES rather than specific entities like (LTER, NEON, CZO). For example, in Australia there is a Terrestrial Ecosystem Research Network (TERN) that has only a single platform with different capabilities. It is important to document these across the globe to be truly an internationally relevant discussion. Regional questions will be answered very differently depending on capabilities, access and funding. To solve global problems we need to go beyond borders. How is this achieved? Needs discussion.

The authors note that "emphasized, discussions with international colleagues are most important" and I would agree, so much so that really that needs to be done as part of

this submission else it is completely biased. This collaboration could/should also come from breakout group or similar at major meetings to enable and capture discussion and debate. It is unclear as to how this has arisen. I suggest that the global community needs to be included now. Also the authors state that "We respond to this request by arguing that NEON can ensure much greater scientific engagement by incorporating more geophysical and biogeophysical aspects of the biogeosciences into their overall design and operations". Again this is a rather flippant statement. What greater engagement exactly? What would this look like? How would it add value to NEON? Isn't this just incorporating CZO into NEON?

Maybe we should embrace the differences you describe between the networks? I think through the paper you have highlighted the differences between the capabilities but perhaps are the strengths. Rather than trying to make them work "much more closely", perhaps again view these as different tools that have their strengths and weaknesses. It will be through our interdisciplinary science approaches (like ESS or CZO etc.) that becomes the enabling factor.

There is a strong focus on 'ecological' and 'earth' sciences but there is more to biogeosciences or ESS. Would be useful to have a more wholistic picture of the interdisciplinary nature of the 'biogeosciences' (which by the way is never defined or scope given).

What are you really asking people to do?? You state that "We call on scientists to accelerate their production of ideas, papers, and proposals for biogeoscience research and education, and to support such research at place-based research sites and across environmental networks." Why in an acceleration needed? Don't we already do this? Not clear on what you are asking from the community and to what end?

The authors talk about scientists and students will provide "The information that will reveal the coverage of data and the data gaps in the three networks". This is a big letdown as I was rather hoping that this paper would tell me this and provide a mechanism

to achieve this. What cross-networks hypotheses are needed? What mechanisms can be used to achieve this USA and globally? Very much left wanting more here and there was a lot of description of the networks to reach a one paragraph statement that does nothing. It certainly doesn't motivate me (which was the aim of the paper). The objective of this paper is to motivate more collaborators to bring the best of the biogeosciences to the LTERs, EONs, and CZOs"

With respect to NEON the authors say that "expanding the scope to include the biogeosciences would involve additional implementation". I question whether that is what is really needed. What is the evidence that there is a lack of biogeoscience in EON's. I would argue that most EON's are lead and driven by biogeoscientists and the community. Again, really what we need here is not biogeosciences injected into everything but rather to utilise the scientific frameworks developed by interdisciplinary science (i.e. ESS and CZS) with appropriate tools to address global problems or answer hypothesis. I can see the attraction of combining CZO's and NEON but I understand that they come from different program areas (and hence funding streams) and serve different core communities. It would have been nice from a design point of view to integrate these right at the start, however, having two capabilities allows independence and the research funds to be distributed not to the usual suspects. There is a danger in having all capabilities rolled into one because then the are operated centrally and you loose buy in from the scientists that need to have a vested interest. Maybe discuss some of these challenges and more systematically?

I found the discussion rather adhoc and biased. 3.1 talks about "biogeosciences and EON/NEON" and 3.2 "The biogeosciences and LTERs and CZOs". Why have one section for EON and another with LTER and CZO combined? Rather get rid of those sections (which are silos to start with) and come up with an integrative framework to look across biogeosciences and the tools and platforms that support them. In all of the discussion there was a push to get CZO into NEON or CZO into LTER. In 3.1 the discussion was a bit of a wish list of CZO measurements that should be made at NEON

sites. In section 3.2 the push was for CZO and drilling at LTER sites. I thought it was an unbalanced discussion and needs to be rewritten. These are all very well and good but why are you asking for all these items for each capability? What specific problem or hypothesis is currently being limited because of the current arrangement? I'd like to see a more systematic review of the measurements required for the biogeosciences and from that the additional tools and measurements that may be needed and where they would be best placed.

Also, in section 3.2 the authors say that "Ways in which we can promote enhancement of the biogeosciences in the research question-driven LTERs and CZOs can be found in the LTER and CZO literature itself." and proceed to describe some examples. These examples showed the conceptual link of the link of geosciences to LTER but the section did not demonstrate any ways to promote the enhancement of biogeosciences in LTERs.

If we added all the missing pieces from each (as you have identified) then don't we end up with three identical capabilities? Is that what we want? Again, maybe the diversity is important.

The last paragraph is really the first and only specific recommendation from the paper. I would like to see a more comprehensive plan for engaging biogeoscientists. Overall paper could have outlined the challenges better and what the opportunities are?

There are some throw away statements about societal issues such as "Such a biogeoscience initiative could help better address a variety of pressing human needs as well. There are growing numbers of biologists and geologists working together on societally important issues." What are these? Cant just say this and leave it you must demonstrate this. These issues can perhaps be a uniting umbrella for biogeoscientists but these need to be articulated.

Last paragraph in the conclusion "good reasons to bring an explicitly biogeosciences initiative to the world's LTERs, EONs, and CZOs". Despite this sounding like a good

thing I don't think the authors have outlined what a 'biogeosciences initiative' is? Again I would argue that 'biogeosciences' is just a name for multidisciplinary research and the principles and frameworks for LTERs, EONs, and CZOs have been developed by communities that have been multidisciplinary. The development of ecosystem ecology, earth system science over the past 2-3 decades have driven these networks so again at the end not sure what the real purpose of this paper was.

Do the Geophysical Unions play a key role in biogeosciences? Discuss?

I found the figures not so relevant, particularly figures 1, 2 and 5 do not add value to the text and I suggest you remove.

―――――――――――――――――――

---

## Author Comment (AC1) · 3 Jul 2018

Authors' responses to review #2. We appreciate the reviews and the editorial work. These comments as a whole pointed to the need for a major rewrite, which we have accomplished. We have taken very seriously both reviews and have substantially rewritten most of the ms. We have internationalized the authorship with scientists from Europe, Mexico, Argentina, Africa, India, and China. The text is internationalized in many ways as well. While we retain much of the historical context of the biogeosciences, which we frankly see as fundamental, we have attempted to make the paper as all encompassing to as many disciplines and audiences as possible. We seem to have been interpreted in our previous version as favoring one environmental network over another. As this was not our message, we have explicitly addressed this fact and have

adopted a much more positive tone throughout. We are on the side of the biogeosciences, not on the side of any one network. By adding the FLUXNET network to the ILTERs, EONs, and CZOs, we believe we have further defused any possibility of being seen to be partisan for one network over another.

We believe you will see that this is much improved ms. We recruited a number of international co-authors and have substantially re-written the text making it a truly international document.

Below see Author responses (led with an A) to reviewer #2 comments (led with an R).

RGlobally, research networks are key to supporting regional and global science. They have not developed equally across the globe and even the progression of networks within a country can be messy. The authors have done a good job at describing the US based networks and the manuscript is generally well written. Therefore, the authors have provided a great platform for discussion and for this reason I think it is important. I am not from one of those networks and I acknowledge that my comments are partly opinion too but hope that it adds to a useful discussion. I will keep my comments general at this time as I think the paper needs major revision.

AWe appreciate these comments from a person "outside" the environmental network communities. As authors, we'd like to think that these networks are sufficiently open to welcome researchers like the reviewer.

RMy first impression is that this is very American centric with the author list and examples of networks used in the paper. Bringing biogeosciences to LTERs, EONs, and CZOs is necessarily an American focus due to the specific infrastructure programs but should be a global one. Is there a better framework here? Or at least better nomenclature? If, as they propose, we are "interested in addressing questions that motivate the worlds research networks" then this requires a global effort and integration across boundaries is necessary. In addition, submission to Biogeosciences is to a global audience and the paper should be relevant. To stay in Biogeosciences I would recommend

that the paper should expand globally to be a real interest and engage our global community rather than assuming that the rest of the world is the same as the USA.

AWe have substantially revised the ms in many ways, but one of the most important was to internationalize the coauthors and the topics discussed in the text. We have also added the FLUXNET network which is a profoundly international network. The revised ms is aimed at a global audience across the full range of the biogeosciences.

RThe objective of the paper was to "motivate more collaborators to bring the best of the biogeosciences to the LTERs, EONs, and CZOs". In my opinion this is a rather odd objective to have for the paper and I don't think it achieved it. The paper suffers from not having a clear problem statement, so it is never clear what the discussion is trying to achieve and so in the end it achieves nothing. The scope and problem need to be well developed with some specific outcomes in mind. For example, what are the pressing problems and puzzles of the world that you allude to and how can they be addressed by integrating different measurement capabilities or networks. What the important global questions that you refer to in the paper? It would be useful to determine these perhaps as an integrating framework – find synergies first.

AWe thought a lot about the best way to revise the objectives and problem statement and have revised this entirely. We lay this out in the 1st paragraph of the new revision. The paragraph reads: "In this paper, we bring the biogeosciences and environmental research networks together by exploring their origins and by asking a simple question: might on-going environmental research networks benefit from a perspective that more explicitly includes the biogeosciences? The specific objectives of this paper are to consider the historical development of the biogeosciences and of environmental research networks, and to use that history to highlight opportunities for the world's environmental research networks to use the biogeosciences to benefit network science itself and to broaden their impacts on the wider sciences and society."

RThere is also no definition of what biogeosciences is and what a is a 'biogeoscience

approach'? Are we talking about Earth system Science or Critical Zone Science?

AWe are writing for a premier journal of the biogeosciences, so there should be good understanding of the important overlap of what we conventionally consider biology and geology. We reiterate our objectives by making a statement about that overlap. "While these networks were founded and grown by remarkably interdisciplinary scientists, the preponderance of expertise and funding streams have tended to gravitate to different networks by discipline: ILTERs and EONs toward ecology and biology, CZOs toward the geosciences, and FLUXNET toward ecophysiology and micrometeorology. While our paper's interest and objective is not to homogenize environmental research networks, we do assert that biogeoscience presents special opportunities for integrating diverse disciplines in ways that will benefit the research networks in advancing science and disseminating their science narratives among scientific communities and the public."

RI agree that these networks are underutilised. Many networks globally struggle with minimal (or no) funding to keep the lights on and funds do not support scientific research. This is a problem particularly for investigators who spend a lot of time running things. Perhaps you could discuss this? The authors ague that "In fact, the core concepts that motivate these networks' operation clearly and substantially overlap (Figure 4), that is, ecology's ecosystem is entirely congruent with Earth science's critical zone". I find this is a cursory observation and the figure simply shows different schematics for each network and does nothing to synthesis the information. There needs much more development of these ideas here. This would be more beneficial if it were a systematic analysis of the operational and conceptual frameworks associated with each. Maybe a table or synthetic diagram. Same with the differences, make a systematic review of the difference and/or gaps. What are the challenges they face?

AWe discuss much more this "congruence" of the core concepts of ecology and Earth science, as ecosystem and critical zone. The FLUXNET's ecosystem is often the aboveground ecosystem and its exchanges with the atmosphere. The ILTER/EON

ecosystem is somewhat deeper, perhaps to 2 m with NEON. However the critical zone ecosystem includes the full depth of biological and weathering influence and water's penetration. The text makes the point that many of the biogeoscience opportunity's are in belowground ecosystems.

R"EONs are surveillance facilities" I don't truly understand the distinctions you are trying to make here. All facilities are surveillance really.

AWe have reworked the descriptive paragraphs (in Section 4) of each network; in fact, these are almost completely revised. Survellance is used in a special way, as in to survele with passive instrumentation that continuously logs environmental data.

RAt the end of the day the networks (or what they provide) are just tools. They are constantly evolving and being reinvented to attract and maintain funding. Really what we want is a scientific framework that is capable of answering these questions BUT then utilises tools in its toolkit. These would include surface observational capability (LTER, NEON, CZO), atmospheric tools, remote sensing and modelling. Shouldn't we also include the humanities? If that is the case don't we already have this is Earth System Science (relevant programs are iLEAPS, GLP, Future Earth). How do the networks help address the big societal questions that they are posing? Are we reinventing the wheel in biogeosciences because we already have ESS? This needs discussion and thought.

AYes, these networks ARE tools, though tools at locations where information is builtup over time, so that we begin to learn a lot about each of the sites. The diversity of each ecosystem and critical zone is quite impressive. And what we learn includes many sciences and yes, even the social sciences and humanities. There is important overlap with iLEAPS, GLP, and Future Earth. We do not think that these projects make the four environmental research networks redundant. It may well be that the approach to science may be quite different in that it might be argued that the iLEAPS-GLP-Future Earth approach starts with the environmental problems and brings science to

find resolutions. Much of the environmental research networks activities focus on the basic science and the environmental dimensions of the science.

RAlso the observational facilities you mention are very specific in the US and are quite different elsewhere in the world so there is a huge need to think about this in terms of CAPABILTIES rather than specific entities like (LTER, NEON, CZO). For example, in Australia there is a Terrestrial Ecosystem Research Network (TERN) that has only a single platform with different capabilities. It is important to document these across the globe to be truly an internationally relevant discussion. Regional questions will be answered very differently depending on capabilities, access and funding. To solve global problems we need to go beyond borders. How is this achieved? Needs discussion.

AThe revision completely opens up this kind of discussion, and we now add FLUXNET with all of its 100s of international flux measurement towers. The capabilities are really incredible, all things considered and by referring to ILTER, EONs, CZOs, and FLUXNET, one hardly has four sets of capabilities. Each of these is diverse with regard to capabilities. Thus CZOs in particular have not protocol for how to design a CZO. The lesson is that each of these systems has inherent diversity in capabilities and that is a fundamental lesson of the paper. We are not after homogenizing the capabilities, just trying to identify the biogeoscience opportunities that might exist across the networks and at particular sites.

RThe authors note that "emphasized, discussions with international colleagues are most important" and I would agree, so much so that really that needs to be done as part of this submission else it is completely biased. This collaboration could/should also come from breakout group or similar at major meetings to enable and capture discussion and debate. It is unclear as to how this has arisen. I suggest that the global community needs to be included now. Also the authors state that "We respond to this request by arguing that NEON can ensure much greater scientific engagement by incorporating more geophysical and biogeophysical aspects of the biogeosciences into their overall design and operations". Again this is a rather flippant statement. What

greater engage- ment exactly? What would this look like? How would it add value to NEON? Isn't this just incorporating CZO into NEON?

AMore than anything, we have completely transformed the manuscript into an international document. This has taken a lot of time and effort. But we think we've done it quite well.

RMaybe we should embrace the differences you describe between the networks? I think through the paper you have highlighted the differences between the capabilities but perhaps are the strengths. Rather than trying to make them work "much more closely", perhaps again view these as different tools that have their strengths and weaknesses. It will be through our interdisciplinary science approaches (like ESS or CZO etc.) that becomes the enabling factor.

AIn point of fact, the more we investigated the differences between networks the more we have come to see that as we say, we do not support a homogenization of the environmental research networks. On the other hand, there are different but important biogeoscience research opportunities that exist within sites and within networks that scientists ought to know about. That is an important part of the paper, in addition of course to a critical thinking of how we gravitate on the basis of discipline.

RThere is a strong focus on 'ecological' and 'earth' sciences but there is more to biogeosciences or ESS. Would be useful to have a more wholistic picture of the interdisciplinary nature of the 'biogeosciences' (which by the way is never defined or scope given).

AYes, there is a wider set of opportunities than the interdisciplinary biogeosciences. We decided to write this for EGU's BGS journal as a good first step. One really can't "solve" problems of the environment within a boundary impermeable to social science and the environmental humanities. Environments have histories afterall! We tried to take a step, rather than run the full marathon that will eventually be run.

RWhat are you really asking people to do?? You state that "We call on scientists to accelerate their production of ideas, papers, and proposals for biogeoscience research and education, and to support such research at place-based research sites and across environmental networks." Why in an acceleration needed? Don't we already do this? Not clear on what you are asking from the community and to what end?

AWe have omitted this argumentation in the new version.

RThe authors talk about scientists and students will provide "The information that will reveal the coverage of data and the data gaps in the three networks". This is a big let-down as I was rather hoping that this paper would tell me this and provide a mechanism to achieve this. What cross-networks hypotheses are needed? What mechanisms can be used to achieve this USA and globally? Very much left wanting more here and there was a lot of description of the networks to reach a one paragraph statement that does nothing. It certainly doesn't motivate me (which was the aim of the paper). The objective of this paper is to motivate more collaborators to bring the best of the biogeosciences to the LTERs, EONs, and CZOs"

AWe've backed away from an ambitious new initiative, etc. We've adopted a more subtle approach in encouraging this interdisciplinarity within existing networks.

RWith respect to NEON the authors say that "expanding the scope to include the bio-geosciences would involve additional implementation". I question whether that is what is really needed. What is the evidence that there is a lack of biogeoscience in EON's. I would argue that most EON's are lead and driven by biogeoscientists and the community. Again, really what we need here is not biogeosciences injected into everything but rather to utilise the scientific frameworks developed by interdisciplinary science (i.e. ESS and CZS) with appropriate tools to address global problems or answer hypothesis. I can see the attraction of combining CZO's and NEON but I understand that they come from different program areas (and hence funding streams) and serve different core communities. It would have been nice from a design point of view to integrate

these right at the start, however, having two capabilities allows independence and the research funds to be distributed not to the usual suspects. There is a danger in having all capabilities rolled into one because then the are operated centrally and you loose buy in from the scientists that need to have a vested interest. Maybe discuss some of these challenges and more systematically?

AMethinks in re-thinking and revising the ms, we've come to a similar direction as seems outlined here. That we do not want to homogenize. On the other hand, there are incredible opportunities and what is discussed in the ms is that while NEON has given us instruments to 2-m depth, FLUXNET experience clearly demonstrates that the water balance can't be closed in most places at 2 m. The same goes for nutrients and other resources, and thus it is quite typical that subsurface characterization is quite superficial. Again, we are not arguing to make all networks the same, as you say there is value in different emphases and capabilities. However, this means there are un- and underexplored opportunities.

RI found the discussion rather adhoc and biased. 3.1 talks about "biogeosciences and EON/NEON" and 3.2 "The biogeosciences and LTERs and CZOs". Why have one section for EON and another with LTER and CZO combined? Rather get rid of those sections (which are silos to start with) and come up with an integrative framework to look across biogeosciences and the tools and platforms that support them. In all of the discussion there was a push to get CZO into NEON or CZO into LTER. In 3.1 the discussion was a bit of a wish list of CZO measurements that should be made at NEON sites. In section 3.2 the push was for CZO and drilling at LTER sites. I thought it was an unbalanced discussion and needs to be rewritten. These are all very well and good but why are you asking for all these items for each capability? What specific problem or hypothesis is currently being limited because of the current arrangement? I'd like to see a more systematic review of the measurements required for the biogeosciences and from that the additional tools and measurements that may be needed and where they would be best placed.

[Figure]

AWe've shifted to a much more practical approach in the new version. We're no longer suggesting there is a big program out there. We've completely revised this Discussion and much more systematically and in a non-siloed fashion opened up some of the opportunities that research managers, researchers, or students might see and find interesting and important.

RAlso, in section 3.2 the authors say that "Ways in which we can promote enhancement of the biogeosciences in the research question-driven LTERs and CZOs can be found in the LTER and CZO literature itself." and proceed to describe some examples. These examples showed the conceptual link of the link of geosciences to LTER but the section did not demonstrate any ways to promote the enhancement of biogeosciences in LTERs.

AWe've upgraded these topics substantialy in the current version. We do not foresee you having this reaction with the revised ms.

RIf we added all the missing pieces from each (as you have identified) then don't we end up with three identical capabilities? Is that what we want? Again, maybe the diversity is important.

ADiversity is good. We are NOT recommending all networks be homogenized. We learn from each other. The networks too often are in their own silos however and it is time to have them start to interact!

RThe last paragraph is really the first and only specific recommendation from the paper. I would like to see a more comprehensive plan for engaging biogeoscientists. Overall paper could have outlined the challenges better and what the opportunities are?

AWe have advanced this paragraph to the front end of the ms, so that it makes clear from the start what we are doing. Again, this is not a big NSF project, capital P, but rather a big decentralized effort on the part of biogeoscientists.

RThere are some throw away statements about societal issues such as "Such a biogeoscience initiative could help better address a variety of pressing human needs as well. There are growing numbers of biologists and geologists working together on societally important issues." What are these? Cant just say this and leave it you must demontrate this. These issues can perhaps be a uniting umbrella for biogeoscientists but these need to be articulated.

AWe've edited the ms very closely.

RLast paragraph in the conclusion "good reasons to bring an explicitly biogeosciences initiative to the world's LTERs, EONs, and CZOs". Despite this sounding like a good thing I don't think the authors have outlined what a 'biogeosciences initiative' is? Again I would argue that 'biogeosciences' is just a name for multidisciplinary research and the principles and frameworks for LTERs, EONs, and CZOs have been developed by communities that have been multidisciplinary. The development of ecosystem ecology, earth system science over the past 2-3 decades have driven these networks so again at the end not sure what the real purpose of this paper was.

AWell that is our challenge, I can see. To convince readers like you that there are major scientific opportunities in these networks!

RDo the Geophysical Unions play a key role in biogeosciences? Discuss?

AThis is far more than the Biogeoscience Section of AGU or EGU. Formal sections are more symptoms than causal determinants which are much more scientific.

RI found the figures not so relevant, particularly figures 1, 2 and 5 do not add value to the text and I suggest you remove.

AThis critique guided our presentation of the six figures to ensure that they really contributed to the ms.

---

## Author Comment (AC2) · 3 Jul 2018

We appreciate the reviews and the editorial work.

We have taken very seriously both reviews and have substantially rewritten most of the ms. We have internationalized the authorship with scientists from Europe, Mexico, Argentina, Africa, India, and China. The text is internationalized in many ways as well. While we retain much of the historical context of the biogeosciences, which we frankly see as fundamental, we have attempted to make the paper as all encompassing to as many disciplines and audiences as possible. We seem to have been interpreted in our previous version as favoring one environmental network over another. As this was not our message, we have explicitly addressed this fact and have adopted a much more

positive tone throughout. We are on the side of the biogeosciences, not on the side of any one network. By adding the FLUXNET network to the ILTERs, EONs, and CZOs, we believe we have further defused any possibility of being seen to be partisan for one network over another.

We believe you will see that this is much improved ms. Because we were not seen to be advancing all long-term environmental research networks (specifically ILTERs, CZOs, EONs, and FLUXNETs), we have recruited a number of international co-authors and substantially re-written the text.

Responses to review #1. Reviewer comments start with R; Authors responses begin with A.

RThe authors present a very important manuscript advocating better integration of Observatories (EONs) and networks (LTER and CZO) through the biogeosciences. I strongly agree with the authors that this advocacy is needed. The topic of this manuscript is timely and pertinent. I greatly appreciate all the hard work that has gone into this crafting this manuscript.

AThanks for the review and for the complements on "a very important manuscript."

RI am very familiar with most all the co-authors, the observatories and networks, the scientific rationale(s), and working in these organizations. I am also quite familiar with the subject matter. I feel I have broad knowledge of this area of research and development, and feel this is a fair and honest review. I feel that if a reviewer has constructive and collegial comments, they should not hide behind anonymity. Hence, I do not wish to remain anonymous, and please consider this review signed, Hank Loescher. At this time, I recommend a major revision.

AFair enough. We went back to the drawing board and have substantially revised the ms. We have given the ms a new tone for sure, making it much more positive throughout.

RThis is a great opportunity to advance Earth system science, but with all due respect to my esteemed colleagues, I think you have missed the point:

AWith all due respect, our opportunity is to advance the biogeosciences and long-term environmental research networks.

RFirst and most importantly, what is being advocated is as much cultural issue in conducting science as anything. I have seen the direct effects of different scientific cultures in numerous, recent meetings with members of the EONS, LTER, CZO and other networks present. In that the authors are well respected members of the ecological community, there is a large responsibility to communicate your message correctly and with an awareness of the cultural and political sensitivities. Advocating change will be best served by communicating your ideas in a way that can be heard by all the respective user communities. As it is written, I find obvious biases in how each of the research structures are described, and this does a disservice to the goals of the paper and to the future user communities. Authors hold onto an old paradigm of how science is best performed (e.g., no faults to the LTER approach), clearly contain their own bias, and do not present a meaningful path forward. There are a lot of misnomers and imprecise statements that also show this bias, and if published as is, would propagate these biases to the user communities. The issues at hand are not about perpetuating the LTER paradigms in the use of 'hypotheses questions' moving forward, but embracing (and owning as a community) new institutionalized approaches (networks and observatories) to challenge our current approaches (get out of our current boxes) and move the science forward with new tools. I do not see that the authors take ownership if this problem.

AAs co-authors are leaders in the ILTERs, CZOs, NEON, and FLUXNET, we have emphasized during the revision that we will minimize "biases" in how the networks were to be presented. The point is NOT to advocate for one approach or another, the point is to get the most out of all environmental research networks! We are earnestly convinced that the biogeosciences offers great opportunities for science and for engaging larger

scientific and public audiences.

RSecond, the authors do not clearly and objectively describe the respective strengths, weakness and complementarity of all these research organizations. I strongly recommend an unvarnished assessment of the attributes and approaches of the respective organizations. The manuscript would be better served if the authors provide specific science themes together with the approaches that can integrate the data and advance our understanding across disciplines, processes, time, and space.

AWe respectfully have not presented a scientific evaluation of the strengths and weaknesses of the networks and do not see that as being in line with our objectives, which are clearly to promote more integrative biogeoscience within the networks.

ROther key issues EONs (NEON) and networks (CZO/LTER) are BOTH question based. But how they are applied (EONs to inform requirements and top down designs) and Networks (bottom up tradition hypothesis testing) differ. Both utilize the same suite of questions, the nuance is HOW they are implemented. And I fully agree that there has to be a structure in place to revisit, revise, and update the EON capabilities against the rubric of frontier science questions.

AWe carefully have reword these descriptions of question-based, hypothesis-based, etc.. We have checked the literature and included new citations of discussions that compare these networks (Lindenmayer, Likens, Franklin) to ensure we describe the networks accurately.

RThe text often jumps from idea to idea, from concept to concept without really discussing the issues or core mechanisms to really bridge disciplines and integrate concepts. Narrative structure needs more synthesis-style of writing.

AWe have reorganized the paper, hopefully making it more integrated. We respectfully disagree about the jumps from idea to idea. We kept this comment in mind in the revision, so that the story better told.

RIf you look at the original planning documents for LTER, they look a heck of a lot more similar to an EON than what their organizational structure and function are today. Why is that? The change in organizational approach of networks (LTER) over time is natural in its evolution/development. Acknowledging this and advocating for change in the context of biogeosciences among networks is very natural–and messy way that we do science. Maybe state as much.

APerhaps the reviewer is correct on the overlap of the contemporary EON-early LTER literature. What has struck the authors is that the early LTER literature was reaching out in space and time, and occasionaly the audience was explicitly the Earth science community. We have cited these examples and one of the figures demonstrates this in spades (Fig. 5) from the classic paper by John Magnuson on lake productivity being controlled by groundwater flow paths through glacial moraines with contrasting carbonate chemistries.

RAuthors do not really take ownership of the issues at hand, or the process of integrative change in EON structures. Rather, they pointing out the shortcomings and advocate the same old, the same old. The only difference in this manuscript is that biogeosciences is being broadly advocated as the integrative theme without any real specifics in how to do this.

ANot sure what the reviewer is driving at here. We have certainly no critiquas on the shortcomings of EONs. That is not part of the paper's mission. specific (nitpicky) comments

L112, what 'new systems analyses'?, best to define it a tad more.

OMITTED.

L122, what 'critical zone scientists'?, best to define it a tad more.

NOT APPLICABLE IN THE REVISION.

L129, the acronym of 'ILTER' is not really defined

DEFINED AND EXPLAINED.

L133, doesn't 'EOS' stand for something? In Greek mythology, Eos is goddess of the dawn.

It is the name of AGU's newspaper. It is correctly spelled "Eos" in the new version.

L135, what 'is a time scale in LTER'?, what is a 'time-scale' in a network? do y'all mean 'site-based LTER science'? or something like that?

We have deleted this discussion from the new version and the new version uses Tom Callahan (1984) to discuss time scales and ecological research.

L141, what 'NEON'?, best to define it a tad more. Peters et al. 2014 provides a working description of 'EONs' and other ecological infrastructures.

Thanks for the suggested citation. We have overhauled our description of EONs and NEON in particular.

L143, what about temporal scales?, re. designed to scale in both time and space. . .

Temporal scales are interesting in EONs as they span the near instantaneous to the decadal.

L144, I think the statement of 'ecological conditions and biodiversity' falls at tad short. (i) does not embody the philosophical approach of cause and effect, and (ii) in the case of NEON there are 7 grand challenge areas that were adopted by the NAS 2001, 2003 reports. Moreover, many would argue that no EON can estimate 'biodiversity' very well. The specific approach and rationale for biodiversity observations have to be taken into account, and towards what end? Moreover, biogeochemistry (that y'all are arguing for) is also being measured in NEON, though maybe not necessarily measuring what individual investigators want, or to a desired fidelity. The EON design criteria includes; (i) to be applicable to a broad user community, (ii) to be considered data product/approach broadly accepted by the user community, (iii) data product/approach not be considered

experimental, and (iv) under pragmatic and fiscal constraints. All EONs have the same design issue. Best to highlight that there are other EONS internationally (just like that noted for iLTER and iCZOs)

While we discuss briefly international EONs including TERN in Australia and mention GEO projects, the paper is not about biology, and thus biodiversity is mentioned briefly. We respectfully do not see ourselves discussing EONs that are designed as biological networks, EONs/BONs. Our field is the biogeosciences.

L146, 'question-driven' implies that 'EONs' are not question driven. This is not true, and does a disservice to the community and emergent culture of integrated research infrastructures. See note above.

We have carefully reworded these statements, and certainly meant no "disservice to the community and emergent culture". Our purpose is the help build community and advance the networks.

L156, while it is nice to see the whole book referenced, (re. Chabbi and Loescher 2017), the point that y'all made was in the chapter; Loescher, H. W., E. Kelly, and R. Lea, 2017 National Ecological Observatory Network: Beginnings, Programmatic and Scientific Challenges, and Ecological Forecasting. In: Terrestrial Ecosystem Research Infrastructures: Challenges and Opportunities. Eds. A. Chabbi, H.W. Loescher. CRC Press, Taylor Francis Group, Boca Raton, FL, pp. 27-48. ISBN 9781498751315. L165, consider '. . .and Earth scientists, alike. . .'

While we appreciate the suggested citation, we no longer make this argument that the environmental networks resemble space telescopes, seismic recorders, and the like.

L167, what does 'best of the biogeosciences' mean? best to be a tad more concise in the writing and a little less arm-wavy.

We omitted these phrases entirely.

L170 the comment '. . .to work across these networks to help solve pressing environmental problems and puzzles.' confuscates the difference between the need to advance basic science and understanding with the need to demonstrate societal benefit, economic value and an applied approach. Which is it? Best to be a tad more concise.

These phrases are omitted and no longer a part of the ms.

L170, the voice and tone. . . reads tad self-serving. The idea to integrate biogeosciences among several science disciplines has been around for a long time. Authors are correct to point out that reductionism plays a part (historically), but also note that a resistance to change current approaches is much more of an issue today. It seems as though there are a lot of issues are raised without fully embracing a synthetic statement or path forward... (i) basic science, developing a are under- standing, and discovery, (ii) applied science toward decision makers (natural resource management), and (iii) policy driven science, all of which have different implication of how science is being done in the context of a network or infrastructure.

We stand behind our use of reductionism in the brief historical description of the evolution of biology and geology as departments, disciplines, and frames for organizing research and education (Section 2, paragraph 2) . Most of the rest of these points not longer pertain to the revised ms as the verbiage and ideas surrounding line 170 has been removed from the ms.

L181, Schimel, D., M. Keller, S. Berukoff, R. Kao, H. W. Loescher, H. Powell, T. Kampe, D. Moore, and W. Gram, 2011. NEON Science Strategy; Enabling continental-scale ecological fore- casting. Pub. NEON Inc., Boulder CO. pp 55. [webpage citation] is a more appropriate reference.

Thank you for the suggested citation. The networks are introduced in an entirely different manner and citation is not as relevant in new version.

L227, why are 'field experiments',Âa ÌĘimportant? What is the philosophical context

that becomes important in your narrative? Someone can say, so what, LTER has experiments?, towards what end?

The conventional reply is that field experiments in many Long Term Ecological Research sites test hypotheses about processes that affect ecological change over a number of years to several decades.

L227 while LTER research embraces different themes, PI based research is question/hypotheses driven, which also can be seen as a limitation, because of its lack of integration among other science that is being done at and among sites regardless of its utility to public policy. See comment above.

We need a variety of approaches to do environmental science well. We need NEON's and TERN's approach, placed-based research sites, modeling, and targeted short-term projects too.

L233, EON's in operations are not 'projects', particularly in the eyes of NSF. They are Large Facilities, or operational Research Infrastructures. Best to change the text to reflect this.

Yes, we agree, NEON is a large facility, though I do not believe we get into these NSF-details in our new version.

L240, it is not a have '30-y vision', it has an NSB approved operational timelime of 30-y. This is a very different thing.

We stand corrected, and omit the entire reference to 30 years.

L235, I strongly disagree with the statement 'EONs are not question-based or hypothesis-testing projects', and it does a large disservice to the user communities. They have been informed by grand challenge questions (from NAS in the case of NEON), and investigator based hypotheses. HOW they use them in the design is different. And I fully agree that there has to be a structure in place to revisit, revise, and update the EON capabilities against the rubric of frontier science questions. See

comment above.

We no longer so explicitly state that "'EONs are not question-based or hypothesis-testing projects."

L237, what is a 'highly controlled measurement'?, not a very concise statement. Do y'all mean measured in the same way across all the sites, with the same sources and magnitude of uncertainty, rigorously QA/QC'd, to assure robust cross site analyses?

We have reworded this point focusing NEON's "control" on protocols and instrumentation. CZOs and LTERs measure pH of water for example, but not with the same instruments and sensors. That is the control NEON exerts over its instruments!

L238, L249, 'intended' sounds a bit arbitrary and argumentative, I would suggest to use 'designed'. L241, not really a network, NSF prefers 'Facility', 'Infrastructure' or Observatory. See comment above.

Ok, but that is a fact of history, as "network" is in the name NEON.

L247, L250-251 it is definitely NOT 'NEON's mission is to analyze and forecast impacts of. . .'. NEON is charged to provide the data to enable an ecological forecasting. NEON is NOT preforming any of the data analyses or forecast ecological processes– that is for the community to do. I am quite surprised that this narrative was crafted this way, given that some of the co-authors are intimately aware of this point.

The authors fully understand these points. It is the community's job to do the analysis and forecasting. The first version was perhaps not written as clearly as it needed to be.

L253, what does '. . .tightly controlled. . .' mean?, all NEON, TERN, SAEON data is open access. . . please be concise in your meaning here.

This is a similar point to the comment made about L237 above. Control is about protocol and instrumentation.

L255, 'short time scales'? Not quite as concise and embracing a narrative that you

could use. If you are discussing these data with an ecosystem scientists, they would potentially think decal scale data is very long, if chatting with micrometeorologists, they would think decadal scales would be infinite! I suggest to qualify this statement as something like . . . 'short time scales when compared to geological timescales that CZO community is accustomed to.', or something towards that effect.

We agree with the reviewer to be more explicit is framing short-term and long-term etc. Our co-authors who are "micromet specialists" and "ecosystem scientists" are certainly interested in far more than is suggested in the comment. Our micromet specialists, ecosystem scientists, and geoscientists have conventional interests about time scales but they are also interested in the full range of time scales as presented in Fig. 6.

L257-60, NEON's current design does little to accommodate change. I do not disagree. But the verbiage is adversarial, rather than engaging.

The new version has NO adversarial verbiage. Please excuse our former writing if you took it as such.

L276, 'NSF' is not defined,

DEFINED.

L278-287, awkward sentences, suggest re-crafting it.

OMISSION IS the best way to edit awkwardly written verbiage!

L290, verb missing?, '. . .pertinent to [understand?] critical zone structure and function.' I do not think a study itself is pertinent to the CZO structure.

Omitted entirely.

L291-298, awkward sentence, suggest re-crafting it.

Omitted entirely. Thank you for suggestion!

L298-305 seems like a laundry list without any real syntheses of why these are important. Moreover, the paragraph begins w/ US CZO and then China, Mexico, France and India are mentioned. Best introduce there is a iCZO network analogous to iLTER.

This "laundry list" is omitted.

L310-313, redundant, re-write or remove

More carefully written. We stand behind this statement and have added a classic citation (Evans 1956).

L314, I disagree. A meteorologist is concerned at the synoptic or orographic time scales. Some CZO sites measure the turbulent exchange (much shorter timescales), hence it would be more concise to state 'micrometeorologist'. L316, are they really 'young', or early career?

"MICROMETIOROLOGY" IT IS, "EARLY CAREER" IT IS. Thank you.

L320, interests? Unclear.

We may disagree. Because ecologists and Earth scientists have many shared interests. This is the basis for our Tansley Essay (2015) on why Tansley's ecosystem is so similar to Earth scientist's "critical zone." We need to capitalize on the fact that our core concepts overlap so much.

L323, 'informational and physical' = good!

Thanks for the compliment!

L323, last clause '...and, expertise across LTERs, ...' seems out of place and a vestige from edits.

Edited out of the ms.

L324-333, seems arm wavy, what is different here than already being done (to play devil's advocate) what is really new here? What is the nitty gritty here? Just saying we need it, is not different from what current scientists do. . .

We've omitted this entirely.

L340, what happened to atmospheric?

It was carelessly not included. We've omitted this line of reasoning.

L390, 'full bloom' is jargon and very odd choice of wording. Suggest crafting the test with more approachable narrative structure.

Omitted this phrase. We've also very substantially revised the narrative.

L392-L395, while there are interesting points here, the text fails to synthesis the core integral concepts that are needed to advance our science. Stating that instantaneous to millennial timescales are addressed, but fails to discuss how this is done and to what end. What is the nitty gritty here?

Thanks for the compliment. We have reorganized and given structure to these arguments. See new Section: 5.1.3

L399, what 'benefits'?, and L402, what 'great opportunities'? just stating so, does not make it so. L418-420, How are these ideas being integrated?, merely stating so does not make reality.

As stated immediately above, we've given a structure to discussing opportunities. Here we are in the new version's Section 5.1.1.

L431, Advocating a call to action to 'research agencies' is parochial at best. Suggest figuring out a different way to articulate this. Rarely does such a statement effect change in the programmatic activities or funding opportunities of an agency.

We agree, and have re-articuled this "call" in a very different manner. We are NOT out to homogenize networks. Diversity of networks may be a good thing. However, networks need to get the most out of their investments and opportunities for interdisciplinary research at individual sites or across networks can be great for the the networks, for science, and for engagement by the public!

L465, no Acknowledgements?

On new page 21.

L469, seems like a verb is missing.

Reworded.

L592, is Josh Schimel, not O Schimel.

Corrected, thanks.
* * *

---

## Author Comment (AC3) · 3 Jul 2018

The ms is much revised.

The two reviews were admittedly critical, but the authorship found the resilience and the perseverance to complete a thorough revision, attempting the address and satisfy nearly all concerns of the reviewers.

The scope of the ms is now throughly international, both in the text and co-authorship, with authors from each of the networks reviewed, and a number from outside the network community. We recruited the participation of colleagues from Europe, Asia, Central and South America, and Africa. The content of the text also has an decidedly international scope.

[Figure]

The tone and narrative presentation are also new and entirely revised, with objective description and evaluation of the networks examined, which now include the ILTER, EON, CZO, and FLUXNET networks. The FLUXNET has been added at the suggestion of a physical scientist and we believe greatly impacts the ms. The point of the paper is to discuss the many biogeoscience opportunities for research that often involve the dynamic structure and processes of the subsurface ecosystem.

Our key message is that: While these networks were founded and grown by remarkably interdisciplinary scientists, the preponderance of expertise and funding have gravitated activities of ILTERs and EONs toward ecology and biology, CZOs toward the geosciences, and FLUXNET toward ecophysiology and micrometeorology. Our point is not to homogenize networks, nor to diminish disciplinary science. Rather, we argue that by more fully incorporating the integration of biology and geology in long-term environmental research networks, scientists can better leverage network assets, keep pace with the ever-changing science of the environment, and engage with larger scientific and public audiences.

We sincerely appreciate your work on this ms very much.

Please also note the supplement to this comment:
https://www.biogeosciences-discuss.net/bg-2018-67/bg-2018-67-AC3-supplement.pdf

[Figure]

**Supplement:**

**Strengthening the biogeosciences in environmental research networks**
Ver. 2.7.18

[revised manuscript text omitted]

---

## Author Response (AR2)

**Responses to Ass. Editor's Request for Minor Revisions**
Daniel Richter
20 July 2018

*We appreciate the work of the editor.  I've included editorial comments here in entirety in plain text, as well as our responses which are in italix.*

The historical context of geoscience and bioscience as diverging disciplines is interesting and the justification for their further re-integration is well-written. I appreciate the attention to detail regarding the benefits and disadvantages of the network approach given high-profile criticisms of many network approaches (e.g. Lindenmayer et al., doi: 10.1016/j.tree.2017.10.008).

*Thank you.  We agree that the historical development of the biogeosciences is of more than of mere "historical interest".*

Note on line 266 that flux towers measure these fluxes, although sometimes they have to be estimated, and of course all measurements include uncertainty.

*We change this statement to read:*
*"FLUXNET (the global network of flux towers that  **measure** land-atmosphere exchanges of energy, water, and carbon)."*

I also wouldn't say that flux-gradient techniques have 'collapsed' as they are still in use, albeit infrequent. (I do very much agree that attention to soils in FLUXNET is cursory at best.)

*I have discussed this comment with co-author Dennis Baldocchi who agreed that "collapse" might have been too strong of a verb.  We substitute "collapse" with "near collapse" as when K theory and flux-gradient techniques were recognized not to be practically applicable, it was all but abandoned.*

*According to Baldocchi, in a 20 July email to Richter, "Eventually all/most of us abandoned use of K theory over forests. it suffered from two problems.  Mixing was so great that gradients were tiny and hard to measure, plus K suffered from non local transport, as shown in later papers by Hogstrom/Bergstrom in Sweden and Shaw/Thurtell over borden forest in Canada and the Raupach papers over Uriarra in Australia."*

The paper would also benefit from a quick check for usage (see for example line 563 and rogue minus sign on line 623).

*I have made "a quick check for usage", and believe we have a clean ms.  I've had my lab mgr read thru the ms as well.  The two examples on lines 565 and 623 have been corrected.*

Note also, as it happens, that there is an interesting study on animal contributions to biogeochemistry in Luquillo (doi: 10.1007/s00442-002-1071-9).

*Thanks for this interesting citation.  I tried to add it to the ms, but the paragraphs in which we discuss animal interaction studies, are directed at bio-geomorphological topics, so I decided not to force the citation into the ms.  I appreciate the citation as it helps make the case for biogeosciences.*

Fig. 2b doesn't come from Hood et al., 2006.

*This has been corrected!  Thank you.*

Please make these rather minor suggested changes and thank you for the comprehensive overhaul of the manuscript in response to the referee comments.

*We have made these changes and you are certainly welcome.*